# Object-Centric Slot Diffusion

**Jindong Jiang**
Rutgers University
jindong.jiang@rutgers.edu

**Fei Deng**
Rutgers University
fei.deng@rutgers.edu

**Gautam Singh**
Rutgers University
singh.gautam@rutgers.edu

**Sungjin Ahn**[*]
KAIST
sungjin.ahn@kaist.ac.kr

## Abstract

The recent success of transformer-based image generative models in object-centric learning highlights the importance of powerful image generators for handling complex scenes. However, despite the high expressiveness of diffusion models in image generation, their integration into object-centric learning remains largely unexplored in this domain. In this paper, we explore the feasibility and potential of integrating diffusion models into object-centric learning and investigate the pros and cons of this approach. We introduce Latent Slot Diffusion (LSD), a novel model that serves dual purposes: it is the first object-centric learning model to replace conventional slot decoders with a latent diffusion model conditioned on object slots, and it is also the first unsupervised compositional conditional diffusion model that operates without the need for supervised annotations like text. Through experiments on various object-centric tasks, including the first application of the FFHQ dataset in this field, we demonstrate that LSD significantly outperforms state-of-the-art transformer-based decoders, particularly in more complex scenes, and exhibits superior unsupervised compositional generation quality. In addition, we conduct a preliminary investigation into the integration of pre-trained diffusion models in LSD and demonstrate its effectiveness in real-world image segmentation and generation. Project page is available at `https://latentslotdiffusion.github.io`

## 1 Introduction

The fundamental structure of the physical world is compositional and modular. While this structure is naturally revealed in some data modalities like language in the form of tokens or words, in other modalities such as images, it is elusive how one may discover this structure. However, this representational compositionality and modularity is essential for various applications that require the systematic manipulation of knowledge pieces to achieve high-level cognitive abilities. This includes reasoning [46, 5], causal inference [68], and out-of-distribution generalization [2, 22].

Object-centric learning [27] aims to discover the latent compositional structure from unstructured observation by learning to bind relevant features, thereby forming useful tokens in an unsupervised way. For images, one of the most popular approaches is to auto-encode the image via the Slot Attention [53] encoder. Slot Attention applies competitive spatial attention to partition the image into separate local areas and then obtain a representation, called a slot, from each area. Then, a decoder generates the image from the slots with the aim of minimizing the reconstruction error. Due to the limited capacity and competition between slots, each slot is encouraged to capture a reusable and compositional entity, such as an object.

---

[*]Correspondence to `sungjin.ahn@kaist.ac.kr`.

37th Conference on Neural Information Processing Systems (NeurIPS 2023).

The primary challenge that remains in the framework of unsupervised object-centric learning is making it work for complex naturalistic scene images. Until recently, most object-centric models have adopted a special type of decoder known as the *mixture decoder*. While a weak slot-wise decoder [81] is predominantly employed in this mixture decoder due to its efficacy in promoting the emergence of object-centric representation in relatively simple scene images, further studies [70, 73] have shown that this strong prior can make handling complex naturalistic scene images more challenging. Contrary to conventional belief, Singh *et al.* [70] recently proposed departing from this low-capacity mixture-decoder approach and suggested to use an expressive transformer-based autoregressive image generator in object-centric learning [70, 73]. It was shown that increasing the decoder capacity is crucial for handling complex and naturalistic scenes in this framework [8, 7, 69, 84, 71].

The success of transformer-based image generative modeling in object-centric learning naturally leads to the question: *can diffusion models, another pillar of modern deep generative modeling known for their highly expressive generation capabilities, also benefit object-centric learning*? Diffusion models [75, 31] are based on a stochastic denoising process and have demonstrated impressive performance in a variety of image generation tasks [61, 58, 65, 14, 63, 66, 56], sometimes surpassing transformer-based autoregressive models. Moreover, diffusion models possess unique modeling capabilities that transformer-based autoregressive models cannot provide [75]. However, despite their potential, the applicability of diffusion models to unsupervised object-centric learning remains largely unexplored. Consequently, it is crucial to examine the feasibility of this approach and to identify the associated benefits and limitations.

In this paper, we address this question by introducing a novel model called Latent Slot Diffusion (LSD). The LSD model can be interpreted from two perspectives. From the perspective of object-centric learning, LSD can be viewed as the first model substituting conventional slot decoders with a conditional latent diffusion model, in which the conditioning is object-centric slots provided by Slot Attention. From the diffusion model perspective, our approach is the first *unsupervised* compositional conditional diffusion model. While traditional conditional diffusion models [61, 63, 65, 52] require supervised annotations, such as a text description of an image for compositional generation, LSD is a diffusion model that enables the construction of such compositional descriptions in terms of visual concepts extracted from images through unsupervised object-centric learning.

In our experiments, we evaluate the proposed model across various object-centric tasks, including unsupervised object segmentation, downstream property prediction, compositional generation, and image editing. We show that the LSD model delivers significantly superior performance compared to the state-of-the-art model, namely, the transformer-based autoregressive generative model. A notable attribute of the proposed model is that LSD's performance advantage over autoregressive transformers increases as the scene complexity increases. In particular, LSD enables exploring, for the first time, the applicability of object-centric model to the FFHQ dataset [41], a collection of high-resolution and high-quality face images that surpasses the generative capabilities of existing object-centric models. Additionally, we discuss the overfitting issue faced by LSD on very simple scene images, such as those in the CLEVR dataset [37], and offer suggestions for addressing this problem.

## 2 Latent Slot Diffusion

In this section, we outline the auto-encoding framework of our proposed model, Latent Slot Diffusion or LSD [2], beginning with our object-centric encoder and subsequently describing our proposed decoder.

### 2.1 Object-Centric Encoder

Given an input image $\mathbf{x} \in \mathbb{R}^{H \times W \times C}$, our object-centric encoder seeks to decompose and represent it as a collection of $N$ vectors or slots $\mathbf{S} \in \mathbb{R}^{N \times D}$, where each slot (denoted as $\mathbf{s}_n \in \mathbb{R}^D$) is encouraged to represent a compositional entity in the image. For this, we adopt Slot Attention, an architecture that is also used in the current state-of-the-art object-centric learning approaches [53, 70, 43]. We now describe how Slot Attention works in our model.

In Slot Attention, we first encode the input image $\mathbf{x}$ as a set of $M$ input features $\mathbf{E} \in \mathbb{R}^{M \times D_{\text{input}}}$ via a backbone network $f_\phi^{\text{backbone}}$, *i.e.*, $\mathbf{E} = f_\phi^{\text{backbone}}(\mathbf{x})$. The network $f_\phi^{\text{backbone}}$ is implemented as a CNN

---

[2]Code will be made available at `https://github.com/JindongJiang/latent-slot-diffusion`

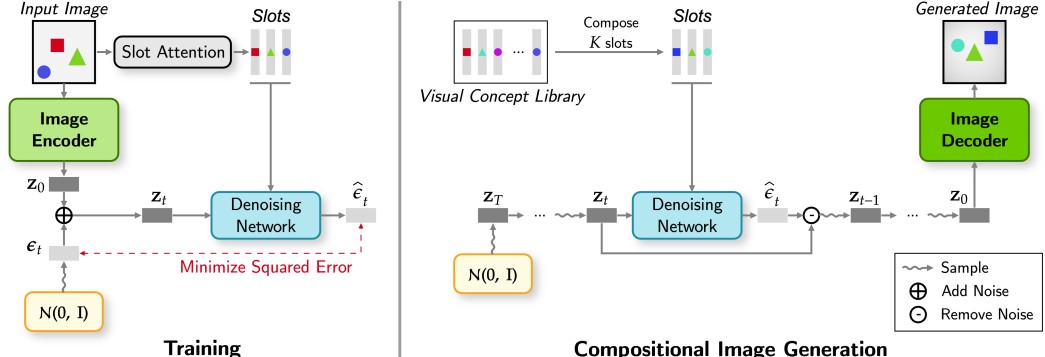

Figure 1: **Method.** *Left:* In training, we encode the given image as image latent and as slots. We then add noise to the image latent and we train a denoising network to predict the noise given the noisy latent and the slots. *Right:* Given the trained model, we can generate novel images by composing a slot-based concept prompt and decoding it using the trained latent slot diffusion decoder.

(detailed in Appendix D.2) whose final output feature map is flattened to form a set. Next, the features in $\mathbf{E}$ are grouped into $N$ spatial groupings and the information in each grouping is aggregated to produce a *slot*. The grouping is achieved via an iterative slot refinement procedure. At the start of the refinement procedure, the slots $\mathbf{S}$ are filled with random Gaussian noise. Then, they are refined via *competitive attention* over the input features, where the $N$ slots act as the queries and the $M$ input features act as the keys and values. The queries and keys undergo a dot-product to produce $N \times M$ attention proportions. Next, on these attention proportions, the softmax function is applied along the axis $N$ to produce attention weights $\mathbf{A}$ that capture the soft assignment of each input feature to a slot. Then, for each $n$, all input features are sum-pooled weighted by their attention weights $\mathbf{A}_{n,1}, \ldots, \mathbf{A}_{n,M}$ to produce an attention readout $\mathbf{u}_n \in \mathbb{R}^D$. These steps can be formally described as follows:

$$\mathbf{A} = \mathtt{softmax}_N \left( \frac{q(\mathbf{S}) \cdot k(\mathbf{E})^T}{\sqrt{D}} \right) \implies \mathbf{A}_{n,m} = \frac{\mathbf{A}_{n,m}}{\sum_{m=1}^{M} \mathbf{A}_{n,m}} \implies \mathbf{u}_n = \sum_{m=1}^{M} v(\mathbf{E}_m) \mathbf{A}_{n,m},$$

where, $q, k, v$ are linear projections that map the slots and input features to a common dimension $D$. Using the bottom-up information captured by the readout $\mathbf{u}_n$, the slots are updated by an RNN as $\mathbf{s}_n = f_\phi^{\mathrm{RNN}}(\mathbf{s}_n, \mathbf{u}_n)$. In practice, competitive attention and RNN update are performed iteratively several times and slots from the last iteration are considered the final slot representation $\mathbf{S}$.

## 2.2 Latent Slot Diffusion Decoder

In this section, we describe our proposed decoding approach called *Latent Slot Diffusion Decoder* or *LSD decoder* for reconstructing the image given the slot representation $\mathbf{S}$. The design of the LSD decoder takes advantage of the recent advances in generative modeling based on diffusion [63, 31]. An overview of our decoding approach is provided in Figure 1.

### 2.2.1 Pre-Trained Image Auto-Encoder

One of the key design components of the LSD decoder is a pre-trained auto-encoder (AE) [21, 63], which provides a way to map an image $\mathbf{x}$ to a lower-dimensional *latent representation* $\mathbf{z}_0$ via an encoder $f_\phi^{\mathrm{AE}}$. This enables LSD to reduce the computational burden of reconstructing high-resolution images by using the lower-dimensional latent $\mathbf{z}_0$ as an intermediate reconstruction target. It also allows us to later obtain the original resolution image without compromising image content and fidelity by decoding the latent $\mathbf{z}_0$ using the AE decoder $g_\theta^{\mathrm{AE}}$. These can be summarized as:

$$\mathbf{z}_0 = f_\phi^{\mathrm{AE}}(\mathbf{x}), \qquad \hat{\mathbf{x}} = g_\theta^{\mathrm{AE}}(\mathbf{z}_0), \qquad \text{where } \mathbf{z}_0 \in \mathbb{R}^{H_{\mathrm{AE}} \times W_{\mathrm{AE}} \times D_{\mathrm{AE}}}, \text{ and } \hat{\mathbf{x}} \in \mathbb{R}^{H \times W \times C}.$$

In our model, we employ an auto-encoder pre-trained on OpenImages [3].

---

[3]We use the pre-trained weights from https://ommer-lab.com/files/latent-diffusion/kl-f8.zip

### 2.2.2 Slot-Conditioned Diffusion

In LSD, we leverage diffusion modeling to reconstruct the image latent $\mathbf{z}_0$ conditioned on the slots $\mathbf{S}$. This modeling approach has been primarily explored in supervised contexts, *e.g.*, for text-to-image generation in Latent Diffusion Models (LDM) [63]. However, unlike LDM, in this work, instead of conditioning the decoder on embeddings of supervised labels, we condition it on slots where the process of obtaining the slots themselves, *i.e.*, Slot Attention, is jointly trained with the decoder without supervision. Following LDM, our decoder works by training a decoding distribution $p_\theta(\mathbf{z}_0|\mathbf{S})$ to maximize the log-likelihood $\log p_\theta(\mathbf{z}_0|\mathbf{S})$ of the image latent $\mathbf{z}_0$ given the slots $\mathbf{S}$. This decoding distribution $p_\theta(\mathbf{z}_0|\mathbf{S})$ is modeled as a $T$-step denoising process:

$$p_\theta(\mathbf{z}_0|\mathbf{S}) = \int p(\mathbf{z}_T) \prod_{t=T,\ldots,1} p_\theta(\mathbf{z}_{t-1}|\mathbf{z}_t, t, \mathbf{S}) \, \mathrm{d}\mathbf{z}_{1:T},$$

where $p(\mathbf{z}_T) = \mathcal{N}(\mathbf{0}, \mathbf{I})$, $p_\theta(\mathbf{z}_{t-1}|\mathbf{z}_t, t, \mathbf{S})$ is a one-step denoising distribution, and $\mathbf{z}_T, \mathbf{z}_{T-1}, \ldots, \mathbf{z}_0$ is a sequence of progressively denoised latents. The one-step denoising distribution $p_\theta(\mathbf{z}_{t-1}|\mathbf{z}_t, t, \mathbf{S})$ is parametrized via a neural network $g_\theta^{\text{LSD}}$ in the following manner:

$$p_\theta(\mathbf{z}_{t-1}|\mathbf{z}_t, t, \mathbf{S}) = \mathcal{N}\left(\frac{1}{\sqrt{\alpha_t}}\left(\mathbf{z}_t - \frac{\beta_t}{\sqrt{1-\bar{\alpha}_t}}\hat{\boldsymbol{\epsilon}}_t\right), \beta_t\mathbf{I}\right), \qquad \text{where } \hat{\boldsymbol{\epsilon}}_t = g_\theta^{\text{LSD}}(\mathbf{z}_t, t, \mathbf{S}),$$

$\beta_1, \ldots, \beta_T$ is a linearly increasing variance schedule, $\alpha_t = 1 - \beta_t$, and $\bar{\alpha}_t = \prod_{i=1}^{t}(1 - \beta_i)$.

**Sampling Procedure.** To sample a $\mathbf{z}_0 \sim p_\theta(\mathbf{z}_0|\mathbf{S})$, we adopt an iterative denoising procedure as in [63, 31]. The sampling process starts with a latent representation $\mathbf{z}_T \sim \mathcal{N}(\mathbf{0}, \mathbf{I})$ filled with random Gaussian noise. Next, conditioned on the slots, we denoise it $T$ times by sampling sequentially from the one-step denoising distribution $\mathbf{z}_{t-1} \sim p_\theta(\mathbf{z}_{t-1}|\mathbf{z}_t, t, \mathbf{S})$ for $t = T, \ldots, 1$. This produces a sequence of latents $\mathbf{z}_T, \mathbf{z}_{T-1}, \ldots, \mathbf{z}_0$ that become progressively cleaner. Finally, $\mathbf{z}_0$ can be considered as the reconstructed latent representation.

**Training Procedure.** Following LDM [63], the training of $p_\theta(\mathbf{z}_0|\mathbf{S})$ can be cast to a simple procedure for training $g_\theta^{\text{LSD}}$ as follows. Given an image $\mathbf{x}$, its slot representation $\mathbf{S}$, and its image latent $\mathbf{z}_0$, we first randomly choose a noise level $t \in \{1, \ldots, T\}$ from a uniform distribution. Given the $t$, we corrupt the clean latent $\mathbf{z}_0$ and obtain a noised latent $\mathbf{z}_t$ as follows:

$$\mathbf{z}_t = \sqrt{\bar{\alpha}_t}\mathbf{z}_0 + \sqrt{1 - \bar{\alpha}_t}\boldsymbol{\epsilon}_t, \qquad \text{where } \boldsymbol{\epsilon}_t \sim \mathcal{N}(\mathbf{0}, \mathbf{I}), \quad \bar{\alpha}_t = \prod_{i=1}^{t}(1 - \beta_i).$$

The noised latent $\mathbf{z}_t$ is then given as input to $g_\theta^{\text{LSD}}$ along with the slots $\mathbf{S}$ and denoising time-step $t$ to predict the noise $\boldsymbol{\epsilon}_t$. The network $g_\theta^{\text{LSD}}$ is trained by minimizing the mean squared error between the predicted noise $\hat{\boldsymbol{\epsilon}}_t$ and the true noise $\boldsymbol{\epsilon}_t$:

$$\hat{\boldsymbol{\epsilon}}_t = g_\theta^{\text{LSD}}(\mathbf{z}_t, t, \mathbf{S}) \qquad \Longrightarrow \qquad \mathcal{L}(\phi, \theta) = ||\hat{\boldsymbol{\epsilon}}_t - \boldsymbol{\epsilon}_t||^2.$$

### 2.2.3 Denoising Network

We implement the denoising network $g_\theta^{\text{LSD}}$ as a variant of the conventional UNet architecture adapted to incorporate slot-conditioning. Our denoising network consists of a stack of $L$ layers where each layer $l$ is a UNet-style CNN layer followed by a slot-conditioned transformer:

$$\tilde{\mathbf{h}}_l = \text{CNN}_\theta^l([\mathbf{h}_{l-1}, \mathbf{h}_{\text{skip}(l)}], t), \qquad \Longrightarrow \qquad \mathbf{h}_l = \text{Transformer}_\theta^l(\tilde{\mathbf{h}}_l + \mathbf{p}_l, \text{cond=}\mathbf{S}),$$

where $\mathbf{h}_0 = \mathbf{z}_t$ is the input, $\mathbf{h}_1, \ldots, \mathbf{h}_{L-1}$ are the hidden states and $\hat{\boldsymbol{\epsilon}}_t = \mathbf{h}_L$ is the output.

**CNN Layers.** Following UNet [64], the convolutional layers $\text{CNN}_\theta^1, \ldots, \text{CNN}_\theta^L$ downsample the feature map via the first $\frac{L}{2}$ layers and then upsample it back to the original resolution via the remaining layers. Following standard UNet, these CNN layers also receive inputs via skip connection from an earlier layer denoted by skip($l$). This network design has also been explored in LDM [63].

**Slot-Conditioned Transformer.** The role of the transformer is to incorporate the information from the slots into the UNet-based denoising process. For this, in each layer $l$, the intermediate feature map $\tilde{\mathbf{h}}_l$ produced by the CNN layer is flattened into a set of features. To this, positional embeddings $\mathbf{p}_l$ are added and the resulting features are provided as input to the transformer. Within the transformer, these features interact with each other and with the slots $\mathbf{S}$, thus incorporating the information from the slots into the denoising process. The transformer output is then reshaped back to a feature map $\mathbf{h}_l$.

# 3 Compositional Image Synthesis

In this section, we describe how a trained LSD model can be used to compose and synthesize novel images. The conventional approach to composing novel images is commonly supervised and rely on text prompts. This involves using words from a vocabulary to create a sentence, which is then given to a text-to-image model to synthesize the desired image [62, 61, 65, 63, 58]. However, in a fully unsupervised setting like ours, we first need to build a library of visual concepts by simply observing a large set of unlabelled images. Then, similarly to composing a sentence prompt using words, we compose a *concept prompt* by selecting concepts from the visual concept library. By providing this concept prompt to the LSD decoder, we can then synthesize a desired novel image. This approach of unsupervised compositional image synthesis was also explored in [70].

**Unsupervised Visual Concept Library.** To build a library of visual concepts from unlabelled images, we first take a large batch of $B$ images $\mathbf{x}_1, \ldots, \mathbf{x}_B$. We then apply slot attention to obtain slots for these images $\mathbf{S}_1, \ldots, \mathbf{S}_B$. Next, we gather all these slots into a single set $\mathcal{S}$ and perform $K$-means on it. The $K$-means procedure assigns each slot in $\mathcal{S}$ to one of the $K$ clusters. We consider the set of slots assigned to a $k$-th cluster as a visual concept library $\mathcal{V}_k$. With $K$ as the number of clusters, this method results in $K$ visual concept libraries $\mathcal{V}_1, \ldots, \mathcal{V}_K$. Our experiments will demonstrate that this basic $K$-means technique can generate semantically meaningful concept libraries. For example, on a dataset of human face images like FFHQ [41], the $K$ libraries correspond to practical concept categories such as hair style, face, clothing, and background.

**Novel Image Synthesis.** Given libraries $\mathcal{V}_1, \ldots, \mathcal{V}_K$, we can compose a concept prompt $\mathbf{S}_{\text{compose}}$ by picking $K$ slots, each from the corresponding $k$-th library, and stacking them together:

$$\mathbf{S}_{\text{compose}} = (\mathbf{s}_1, \ldots, \mathbf{s}_K), \qquad \text{where } \mathbf{s}_k \sim \text{Uniform}(\mathcal{V}_k)$$

We then give the composed prompt $\mathbf{S}_{\text{compose}}$ to the LSD decoder to generate the image latent: $\mathbf{z}_{\text{compose}} \sim p_\theta(\mathbf{z}_0 | \mathbf{S}_{\text{compose}})$. Applying the image decoder on $\mathbf{z}_{\text{compose}}$ generates the desired novel scene image $\mathbf{x}_{\text{compose}} = g_\theta^{\text{AE}}(\mathbf{z}_{\text{compose}})$. For instance, in the FFHQ dataset, the concept prompt can be a collection of chosen hair style, face, clothing, and background. Decoding this prompt would generate a face image that conforms to this prompt.

# 4 Related Work

**Unsupervised Object-Centric Learning.** Object-centric representation learning aims to decompose multi-object scenes into meaningful object entities. A common approach to this is by auto-encoding. In this line, the focus has been to design an appropriate decoder that supports good decomposition. The most widely used decoders include the mixture-decoder [6, 25, 53, 26, 19, 78, 1, 17, 18, 39, 89, 80], spatial transformer decoder [20, 12, 50, 35, 13, 9, 36, 49, 72, 86], Neural Radiance Fields (NeRF) [77, 83, 74], transformer decoder [70, 73, 71, 84, 7, 67, 23], energy-based models [16] and complex-valued functions [55]. Among these, the mixture decoder has been shown to struggle on complex scene images [70, 73] while the spatial transformer decoder requires careful tuning of hyperparameters which limits their applicability in realistic scenes. Neural Radiance Fields require camera poses, and the learning setting involves 3D scenes unlike ours. Another line of work seeks object-centric scene decomposition without reconstruction. This includes works such as [69, 82, 28, 79, 3, 54, 87]. However, unlike ours, these approaches lack the ability to generate images. Preceding object-centric learning, several works pursued disentanglement within a single-vector representation of the scene and use it for compositional image generation [11, 30, 42, 45, 10]. However, lacking spatial binding mechanisms, these methods struggle in multi-object scenes [25] unlike ours.

**Diffusion Models.** Diffusion models (DMs) are a recent class of generative models that can produce high-quality images by reversing a stochastic process that gradually adds noise to an image [31, 76]. DMs have been applied to various tasks in computer vision, such as class-conditioned generation, text-to-image generation, image editing, super-resolution, and inpainting [14, 33, 61, 58, 65, 56, 32, 66]. Recently, [63] proposed Latent Diffusion Models (LDM). By virtue of operating on a low-dimensional latent space, LDM reduces the computational demands of DMs significantly. [51] introduced a method for multi-object scene generation by combining signals from multiple text-conditioned denoising networks. However, to achieve controllable generation, many of these existing DMs require additional labels such as text descriptions to train and control the generation process. In contrast, our model can generate images compositionally using object concepts directly extracted from images. Moreover,

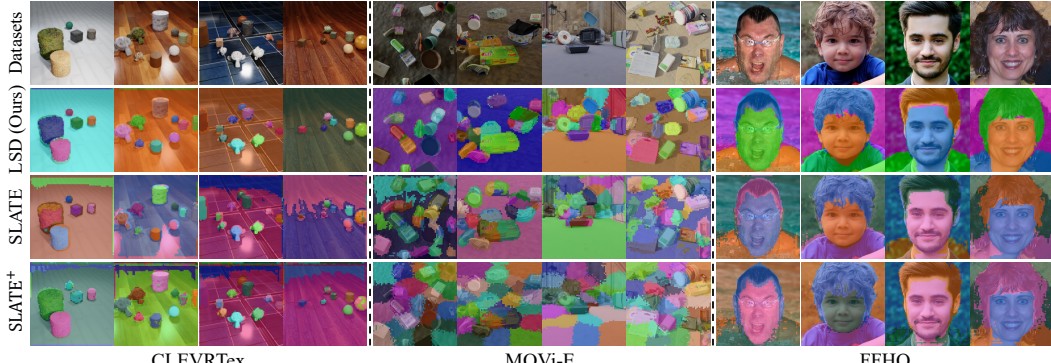

Figure 2: **Visualization of Unsupervised Object Segmentation.** We show visualizations of predicted segments on CLEVR, CLEVRTex, MOVi-E, and FFHQ datasets.

DMs have also been used for representation learning. [4] demonstrated that the intermediate features of a learned DM are useful representations for label-efficient semantic segmentation. [60] introduced an image encoder that compresses an image into a latent representation, jointly trained with the denoising objective. Nevertheless, these representations are unstructured and not modular like ours.

**Concurrent Research.** A concurrent study in [85] also explores a similar idea of object-centric learning using DMs and its downstream applications such as the image editing tasks akin to our experiments. Just like LSD, they employ the Latent Diffusion Model (LDM) as the slot decoder. However, unlike our method, [85] independently train a latent encoder for each dataset, whereas LSD employs a pre-trained latent encoder shared across all input data.

## 5    Experiments

We extensively evaluate our proposed Latent Slot Diffusion (LSD) model on various object-centric tasks, including unsupervised object segmentation, downstream property prediction, compositional generation, and image editing. As will be shown, our model demonstrates substantial improvements over the state-of-the-art on multiple challenging datasets with complex texture and background, including FFHQ [41] which has been beyond the generative capability of object-centric models. In addition, we also include a preliminary exploration into the potential of LSD using pre-trained diffusion models, discussing its performance in both multi-object datasets and unconstrained real-world images in Section 5.4 and Appendix C.

**Datasets.** We evaluate our model on five datasets. Four of them are synthetic multi-object datasets— CLEVR [37], CLEVRTex [40], MOVi-C, MOVi-E [24]. They present increasing levels of difficulty— CLEVRTex adds texture to objects and backgrounds, MOVi-C uses more complex objects and natural backgrounds, and MOVi-E contains large numbers of objects (up to 23) per scene. Furthermore, we explore the applicability of object-centric models to FFHQ [41], a dataset of high-quality face images. Unlike previous works in this line that only investigate low-resolution images, *e.g.*, up to $128 \times 128$, we use a resolution of $256 \times 256$ for all datasets in our experiments.

**Baselines.** We compare our model against SLATE, the state-of-the-art object-centric learning and unsupervised compositional image generation approach. We use its improved version [73], which is more robust in complex scenes. For a fair comparison with LSD that leverages pre-trained auto-encoder, we also develop a variant of SLATE denoted as SLATE$^+$, where its low-capacity dVAE [70] is replaced with a pre-trained VQGAN model [21]. For all models in this work, we use OpenImages-pretrained auto-encoders [63].

### 5.1    Object-Centric Representation Learning

In line with previous work [53], we use unsupervised object segmentation and downstream property prediction to evaluate the object-centric learning capability and representation quality. Within each dataset, all models we compare use the same number of slots.

Table 1: **Segmentation Performance and Representation Quality.** *Left:* We evaluate the segmentation quality and report mBO, mIoU, and FG-ARI scores across various datasets and baselines. *Right:* We measure the representation quality by learning a probe to predict object properties given frozen slots. For position and 3D bounding box, we report MSE. For shape, material, and category, we report accuracy. The results are averaged over three random seeds, and the complete table, including standard deviations, can be found in Figure 5 in appendix.

(a) CLEVRTex

| Segmentation | SLATE | SLATE$^+$ | LSD (Ours) | Representation | SLATE | SLATE$^+$ | LSD (Ours) |
|---|---|---|---|---|---|---|---|
| mBO ($\uparrow$) | 50.88 | 54.90 | **63.93** | Shape ($\uparrow$) | 74.24 | 71.63 | **80.23** |
| mIoU ($\uparrow$) | 49.54 | 52.96 | **62.52** | Material ($\uparrow$) | 69.73 | 63.61 | **75.56** |
| FG-ARI ($\uparrow$) | 44.19 | **70.71** | 64.41 | Position ($\downarrow$) | 1.28 | 1.26 | **1.13** |

(b) MOVi-C

| Segmentation | SLATE | SLATE$^+$ | LSD (Ours) | Representation | SLATE | SLATE$^+$ | LSD (Ours) |
|---|---|---|---|---|---|---|---|
| mBO ($\uparrow$) | 39.37 | 38.17 | **45.57** | Position ($\downarrow$) | 1.37 | 1.28 | **1.14** |
| mIoU ($\uparrow$) | 37.75 | 36.44 | **44.19** | 3D B-Box ($\downarrow$) | 1.48 | **1.44** | **1.44** |
| FG-ARI ($\uparrow$) | 49.54 | **52.04** | 51.98 | Category ($\uparrow$) | 42.45 | 45.32 | **46.11** |

(c) MOVi-E

| Segmentation | SLATE | SLATE$^+$ | LSD (Ours) | Representation | SLATE | SLATE$^+$ | LSD (Ours) |
|---|---|---|---|---|---|---|---|
| mBO ($\uparrow$) | 30.17 | 22.17 | **38.96** | Position ($\downarrow$) | 2.09 | 2.15 | **1.85** |
| mIoU ($\uparrow$) | 28.59 | 20.63 | **37.64** | 3D B-Box ($\downarrow$) | 3.36 | 3.37 | **2.94** |
| FG-ARI ($\uparrow$) | 46.06 | 45.25 | **52.17** | Category ($\uparrow$) | 38.93 | 38.00 | **42.96** |

**Unsupervised Object Segmentation.** Following [53, 69], to measure segmentation quality, we report the foreground adjusted rand index (FG-ARI), the mean intersection over union (mIoU), and the mean best overlap (mBO). These metrics are computed based on the attention masks generated by Slot Attention. Specifically, we utilize the attention weights described in Section 2.1, where attention weights $\mathbf{A}$ are computed as $\mathbf{A} = \underset{N}{\mathrm{softmax}} \left( \frac{q(\mathbf{S}) \cdot k(\mathbf{E})^T}{\sqrt{D}} \right)$.

Our results in Table 1 suggest that LSD is particularly strong in visually complex scenes. For example, on the most challenging MOVi-E dataset, LSD outperforms the strongest baseline by $>8\%$ in mBO, $>9\%$ in mIoU, and $>6\%$ in FG-ARI. On CLEVRTex and MOVi-C featuring complex textures, LSD also achieves $>9\%$ and $>6\%$ gain respectively, in both mBO and mIoU. We visually show the superior segmentation quality of LSD in Figure 2. LSD obtains tighter object boundaries, less object splitting, and cleaner background segmentation. The advantages are most noticeable on CLEVRTex and MOVi-E, where the baselines over-segment the objects more frequently, or exhibit a common failure mode that divides images into approximately uniformly distributed block masks.

We also note that the FG-ARI score is sometimes not an ideal metric for evaluating segmentation quality, because it only considers the foreground pixels and disregards the correctness of the mask shape, as also highlighted by [19, 40]. In the MOVi-E experiment, even though SLATE$^+$ partitions the images arbitrarily into uniform patches, as shown in Figure 2, it still achieves a FG-ARI score comparable to SLATE, which produces significantly more reasonable object masks. This highlights the need for additional metrics, such as the mIoU score that we also measure for evaluating the segmentation quality in this work.

**Downstream Property Prediction.** Following [15, 53], we evaluate the quality of the learned object-centric representations through downstream property prediction. Specifically, for each property, we train a network to predict the property given a frozen slot representation as input. The correspondence between the slot and its true label is determined by Hungarian matching [44] using masks. We use linear heads and 2-layer MLP as the prediction networks for discrete and continuous values, respectively. We report prediction accuracy for discrete properties (*e.g.*, shape and material), and mean squared error for continuous properties (*e.g.*, position).

As shown in Table 1, LSD is competitive with or better than the strongest baseline on CLEVRTex, MOVi-C, and MOVi-E, which is consistent with our finding in unsupervised object segmentation that

Table 2: **Compositional Image Generation.** On various datasets, we generate images using object slots randomly sampled from the dataset. We compare the fidelity of the generated images via the FID score. The lower the FID score, the better the fidelity. We find that our model produces the best fidelity scores compared to the baselines.

| FID ↓ | SLATE | SLATE$^+$ | LSD (Ours) |
|---|---|---|---|
| CLEVRTex | 105.83 | 69.23 | **29.53** |
| MOVi-C | 170.83 | 148.27 | **69.12** |
| MOVi-E | 169.32 | 126.51 | **64.76** |
| FFHQ | 112.38 | 98.76 | **27.83** |

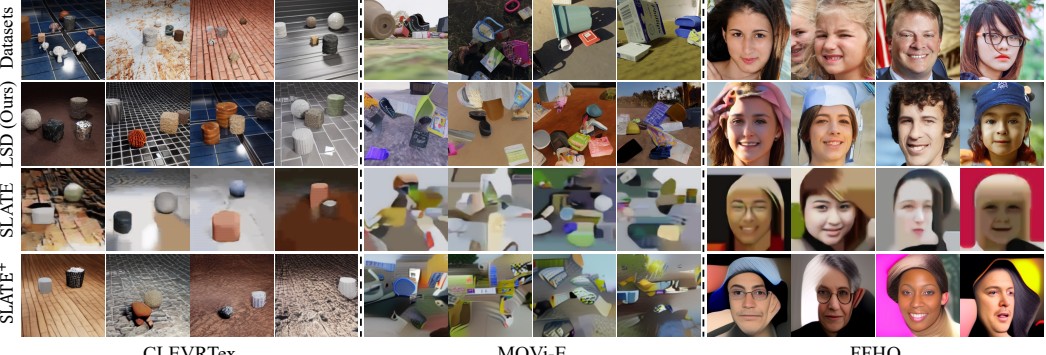

Figure 3: **Compositional Generation Samples.** We qualitatively compare the quality of image samples generated by our model with the baselines. We observe that LSD provides significantly higher fidelity and more clear details compared to the other methods.

LSD shines in visually complex scenes. For example, LSD obtains >4% gain in predicting object category on MOVi-E and object material on CLEVRTex.

## 5.2 Compositional Generation with Visual Concept Library

Like text-to-image generative models, LSD is able to take unseen slot-based prompts at test time and compose new images. Unlike text-to-image models, however, LSD obtains the compositional generation ability solely from images without relying on additional supervision like text inputs. As described in Section 3, we first build a concept library. Then, we sample one slot representation from each visual concept library and concatenate them into a sequence to form a slot-based prompt. We then feed the slot-based prompts to the LSD decoder to generate the images. This produces scenes with novel object layouts and faces with unseen attribute combinations.

We report in Table 2 the FID score [29] as a measure of the compositional generation quality. Following standard practice [14], we compute the FID score using 2K generated images and the full training dataset. Across all datasets, LSD achieves significantly better FID scores than SLATE and SLATE$^+$. We further demonstrate the superior compositional generation quality of LSD in Figure 3. We observe that on the more challenging MOVi-E and FFHQ datasets, LSD generates images with substantially more clear details and better coherence. In contrast, SLATE is limited by its decoder and produces images with missing details in the objects or human faces. While the details can be improved by the pre-trained auto-encoder in SLATE$^+$, some samples still exhibit severe distortions.

## 5.3 Slot-Based Image Editing

In addition to generating new images from randomly sampled slot-based prompts, LSD also allows editing existing images by directly modifying their slot representations. Our experiments showcase LSD's capability in slot-based image editing, including object removal, single-object segmentation, object insertion, background extraction, and background swapping. Remarkably, we demonstrate, for the first time in object-centric generative models, the ability to perform face editing in real-world images, as illustrated in Figure 4.

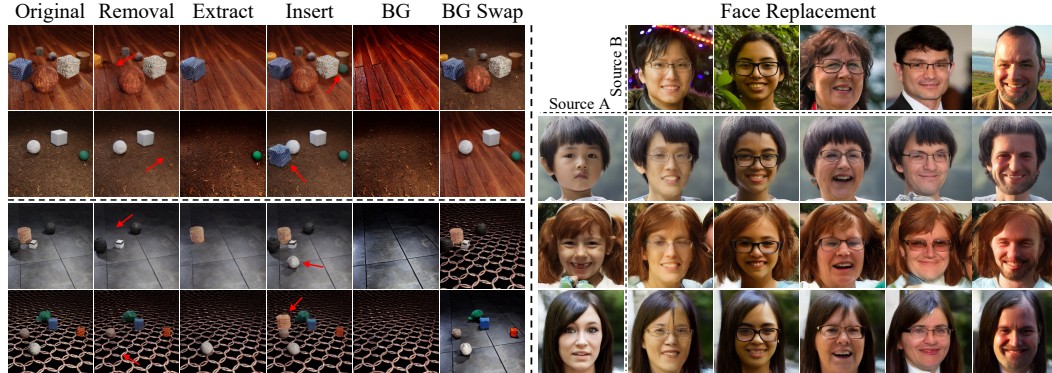

Figure 4: **Slot-Based Image Editing.** *Left:* Our model demonstrates image editing capabilities, including object removal, extraction, insertion, background extraction, and swapping. *Right:* In the FFHQ dataset, we showcase face replacement by combining face slots from Source-B images with hairstyle, clothing, and background slots from Source-A images.

We perform slot manipulation on the CLEVRTex dataset, including object removal, single-object extraction, object insertion, background extraction, and background swapping. To remove an object or extract the background, we discard the relevant slot or all object slots, respectively. Our findings indicate that with near-perfect object decomposition, removing an object can be achieved by simply discarding its slot. Additionally, the background image can be rendered from a single background slot, even without single-slot conditioning during training. In single-object extraction, we render an object with the same background using the respective object and background slots. For object insertion and background swapping, we split the image into top and bottom pairs (Figure 4 *left*) and introduce a slot from another image or swap background slots. This demonstrates that an image's background can be entirely altered while preserving its original objects.

We further explore face replacement on the FFHQ dataset. LSD decomposes each image into four slots, corresponding to face, hairstyle, clothing, and background. Our results show that by replacing the face slots of the images, we are able to coherently change the image while maintaining the hairstyle, clothing, and background. The resulting images look realistic, suggesting that LSD can effectively blend various attributes even when given novel combinations.

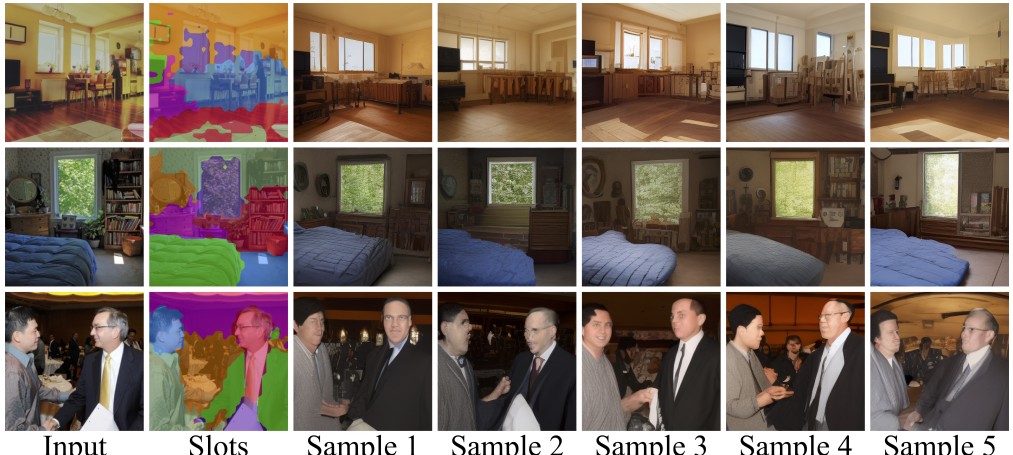

Figure 5: **Unsupervised Object-Centric Learning and Conditional Generation on Real-World Images with Pre-Trained Diffusion Models.** Each image sample is conditioned on the same set of learned slots and with a different initial noise map in the denoising process. We use cfg = 1.3 for all samples. Additional details are available in Appendix C.

### 5.4 Pre-Trained Diffusion Models and Real-World Object-Centric Learning.

In recent studies of DMs, much attention has been given to controllable image generation using large-scale pre-trained diffusion models such as Stable Diffusion [47, 57, 88]. One intriguing question is whether such pre-trained diffusion models can facilitate the learning of object-centric representation. In this regard, we present a preliminary investigation in Appendix C, where we test two LSD variants utilizing the pre-trained Stable Diffusion. Our findings indicate that (1) although the pre-trained Stable Diffusion model has not been trained on any multi-object datasets, it still provides a learning signal for reasonable object-centric learning, (2) with the strong generative capabilities of pre-trained DMs, LSD can scale up object-centric learning to real-world images; however, we also note the part-whole ambiguity issue in slot-based object learning in real-world scenarios, and (3) remarkably, the propose LSD variant achieves conditional generation with unconstrained real-world objects by applying classifier-free guidance [33] to the learned slot, which also provide a visualization of the semantic information captured in the learned representations. We show some image samples in Figure 5, please refer to Appendix C for additional details. We believe these preliminary attempts to be valuable explorations toward large-scale object-centric learning and hope our finding will inspire future research in this area.

## 6 Discussion

**LSD on Simple Images.** While LSD shows significant gains in complex naturalistic scenes, our preliminary investigation suggests a somewhat diminished performance in terms of segmentation and representation when the dataset consists of only visually simple and monotonous scene images like CLEVR. We conjecture that LSD may suffer from an overfitting problem when its high expressiveness encounters visually simple images. To address this issue, additional complexities can be introduced during training, such as integrating images from other datasets. In Appendix B, we offer a more detailed analysis of this challenge using the CLEVR dataset and introduce a training scheme that includes data samples from CLEVRTex to alleviate the problem. Our findings indicate that by integrating greater complexities into the training set, LSD can be effectively employed on simple datasets as well.

**Computation Requirement.** We compare the computation requirement of LSD and the baseline models in the CLEVRTex setting. LSD requires approximately 32 GB of training memory, compared to 31 GB and 36 GB for SLATE and SLATE$^+$, respectively. Notably, LSD does not require much higher memory consumption than regular unconditional latent diffusion, which takes 28 GB under the same setting. We train LSD on 2 NVIDIA RTX 6000 GPUs for 4.5 days, while SLATE and SLATE$^+$ are trained in 1 day and 2.7 days using the same GPU setup. For test-time generation, LSD takes 50.7 seconds to generate 50 images, while SLATE and SLATE$^+$ takes 86.1 and 87.6 seconds, respectively.

## 7 Conclusion

The integration of diffusion models into unsupervised object-centric learning has been largely unexplored, but holds significant potential. To assess the feasibility of this approach and to identify its strengths and weaknesses, we introduced the Latent Slot Diffusion model in this study, which serves two purposes: (1) the first model integrating diffusion models into unsupervised object-centric learning, and also (2) the first unsupervised compositional diffusion model which does not require supervised annotations like text. Our key findings indicate that the proposed model surpasses state-of-the-art transformer-based object-centric models in multiple tasks, with its performance advantage increasing as image complexity grows. Furthermore, the unsupervised compositional generation quality of our model exceeds that of transformer-based counterparts. Finally, we also introduce LSD variants that utilize pre-train Diffusion Models and demonstrate their ability in real-world object-centric learning and conditional generation with real-world objects. We believe that this work represents a significant advancement in object-centric learning towards dealing with complex naturalistic images.

## Acknowledgements

This work is supported by Young Researcher Program (No. 2022R1C1C1009443) through the National Research Foundation of Korea (NRF) funded by the Ministry of Science and ICT. The GPU computing used for this work is partly supported by KI Cloud of Division of National Supercomputing Center, Korea Institute of Science and Technology Information (KISTI). We would like to thank Junmo Cho for managing of specific experiments, Yi-Fu Wu for his contributions to the initial draft of this paper, and Ligong Han and Yuxiao Chen for their valuable feedback on the manuscript.

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

# A    Limitations and Broader Impact

**Limitations** LSD faces several limitations common to unsupervised object-centric representation learning. First, it struggles with part-whole ambiguity, meaning it has trouble distinguishing between individual parts of an object and the whole object itself, especially when objects have complex textures in real-world images. Second, the quality and level of detail in LSD's segmentation results are sensitive to the number of slots it uses. If there are too few slots compared to the actual objects in an image, the model tends to produce generalized masks instead of distinct object segments. Conversely, when there are too many slots, the model tends to over-segment the image, splitting objects into multiple parts. This sensitivity to slot numbers is a problem in scenarios where we don't know the exact number of objects in advance. Finally, even with a strong Diffusion decoder, LSD struggles when dealing with real-world images that have highly diverse object appearances. To address this final limitation, we conduct a preliminary exploration in Appendix C, where we propose two LSD variants and demonstrate their effectiveness in handling real-world data.

**Broader Impact** The proposed Latent Slot Diffusion (LSD) model is a generative model for object-centric representation learning. The main potential negative social impacts is associated with its application in image editing and novel image synthesis. By enabling the extraction of object-centric or component-based representations from images, LSD allows for the generation of entirely new scenes with previously unseen configurations. This capability raises concerns related to image manipulation and privacy. For example, the capability to synthesize new human faces introduces potential privacy issues like deepfake generation and identity theft. Moreover, this ability could potentially cause digital misinformation, as it can generate and manipulate images with high accuracy in object-based level, making it increasingly difficult to distinguish between real and fake or edited visual content. Additionally, the image composition task gives rise to concerns about ethnic bias, ageism, or other forms of misrepresentation. This is due to the uniform sampling process of the learned slots within the same cluster and their independence across clusters. This can result in two significant problems: (1) the sampled distribution may reflect the biases present in the collected data, potentially leading to ethnic, age, or other population biases, and (2) when performing slot swapping for applications such as face replacement, mismatches in gender, age, and skin tone between existing parts and introduced parts may occur. Therefore, further application of this method should be proceeded under strict ethical guidelines and regulation.

# B    LSD on Simple Images

In the paper, we demonstrate that LSD achieves considerable performance improvements for complex images. However, our early experiments reveal that when applied to visually simplistic datasets like CLEVR, LSD exhibits reduced performance in terms of segmentation and representation quality, as illustrated in Table 3. To gain further insights into these results, we provide visualizations of the segmentation outputs in Figure 6.

As illustrated by the LSD samples in Figure 6, certain parts of the background, such as the upper regions of the images, are split and assigned to multiple object slots. This behavior leads to information mixing between objects and the background, which substantially reduces the performance of slot segmentation and representation. We conjecture that this issue may stem from overfitting in the model. Due to the simplicity of the texture and the absence of objects in these areas, the powerful decoder of LSD can easily memorize the rendering process of those regions independently of the slot-conditioning. As a result, it does not provide a sufficient training signal for the encoder to group those background parts into a meaningful slot.

In light of these findings, we tested a straightforward approach to address this problem. We propose introducing additional complexity to the background by incorporating data samples from the CLEVRTex dataset into the training process. By integrating the CLEVRTex samples, we introduce 60 different background textures, which makes implicit memorization difficult. This effectively provides the learning signal to capture the background component into the slot-conditioning. The results of this approach, labeled as LSD(Mix), can be seen in Figure 6. As clearly illustrated, including CLEVRTex data samples in the training results in cleaner background segmentation and alleviates the information mixing issue. As a result, LSD(Mix) achieves approximately 22% gains of improvement in both mIoU and mBO compared to the original model trained solely on CLEVR samples, as shown in Table 3. However, we also see that the FID score indicated a decrease in the generation quality of

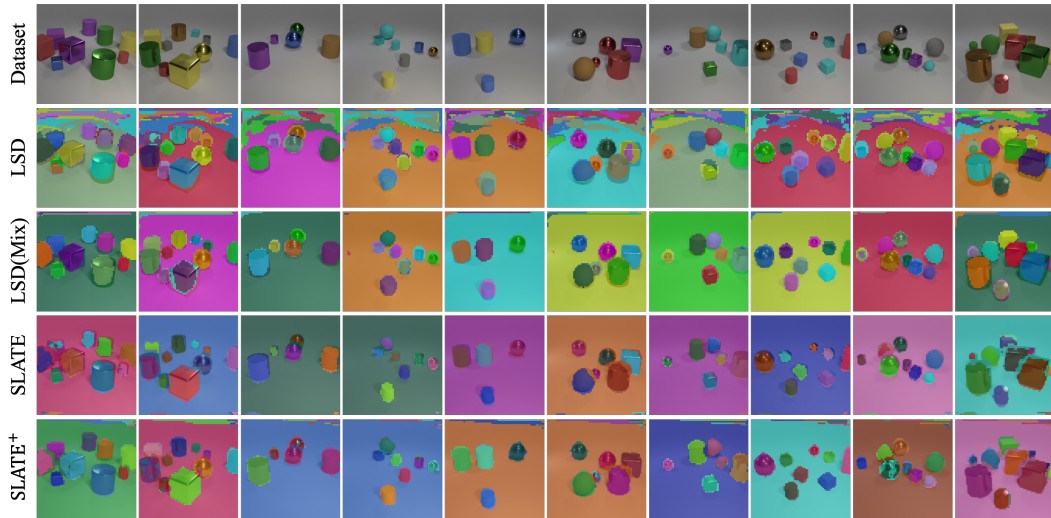

Figure 6: **Visualization of Unsupervised Object Segmentation on CLEVR dataset.** The LSD(Mix) variant corresponds to the model trained on both CLEVR and CLEVRTex training images.

LSD(Mix) compared to the original LSD, likely due to the increased demand of model capacity to model both datasets. Overall, this experiment confirms that introducing complexity into the training dataset can mitigate the overfitting problem and allow LSD to effectively perform representation learning on simple datasets.

## C    LSD with Pre-Trained Diffusion Models

One of the main focuses of recent research in diffusion models have been on adapting pre-trained models for additional control signals [47, 57, 88]. Specifically, these studies propose the integration of a learnable adapter network into large-scale pre-trained diffusion models, such as Stable Diffusion, to enable controllable generation based on various types of conditioning such as sketches, semantic layouts, and depth maps. An inherent advantage of these approaches is their ability to bypass the resource-intensive processes of training the entire generative model, while still enabling the production of high-quality controllable images through minimal training of the adapter network.

We note that such a model design can also be utilized for slot-based conditioning. However, unlike existing works that primarily focus on learning the adapter to integrate the provided conditioning into the pre-trained diffusion model, our focus lies in training an adapter to achieve two objectives simultaneously: (1) we aim to learn how to extract object representations (slots) from input images in an unsupervised manner, and (2) we aim to learn how to incorporate these slot representations into the pre-trained diffusion model to facilitate slot-based conditional generation. This leads to an intriguing question: can learning through such a model, which has not been specifically trained on multi-object

Table 3: **Quantitative Evaluation on CLEVR dataset.** We evalute the models by reporting mBO, mIoU and FG-ARI scores for segmentation quality, property prediction scores for representation quality, and FID scores for compositional generation quality.

|  | SLATE | SLATE$^+$ | LSD | LSD (Mix) |
|---|---|---|---|---|
| mBO ($\uparrow$) | 64.86 | **67.42** | 38.49 | 61.09 |
| mIoU ($\uparrow$) | 63.96 | **66.62** | 37.49 | 59.76 |
| FG-ARI ($\uparrow$) | 69.20 | **88.56** | 76.32 | 76.30 |
| Shape ($\uparrow$) | 95.66 | **95.70** | 75.16 | 80.48 |
| Material ($\uparrow$) | 97.62 | **97.96** | 94.21 | 96.24 |
| Position ($\downarrow$) | 0.59 | **0.51** | 0.86 | 0.99 |
| FID ($\downarrow$) | 52.96 | 20.93 | **16.22** | 19.45 |

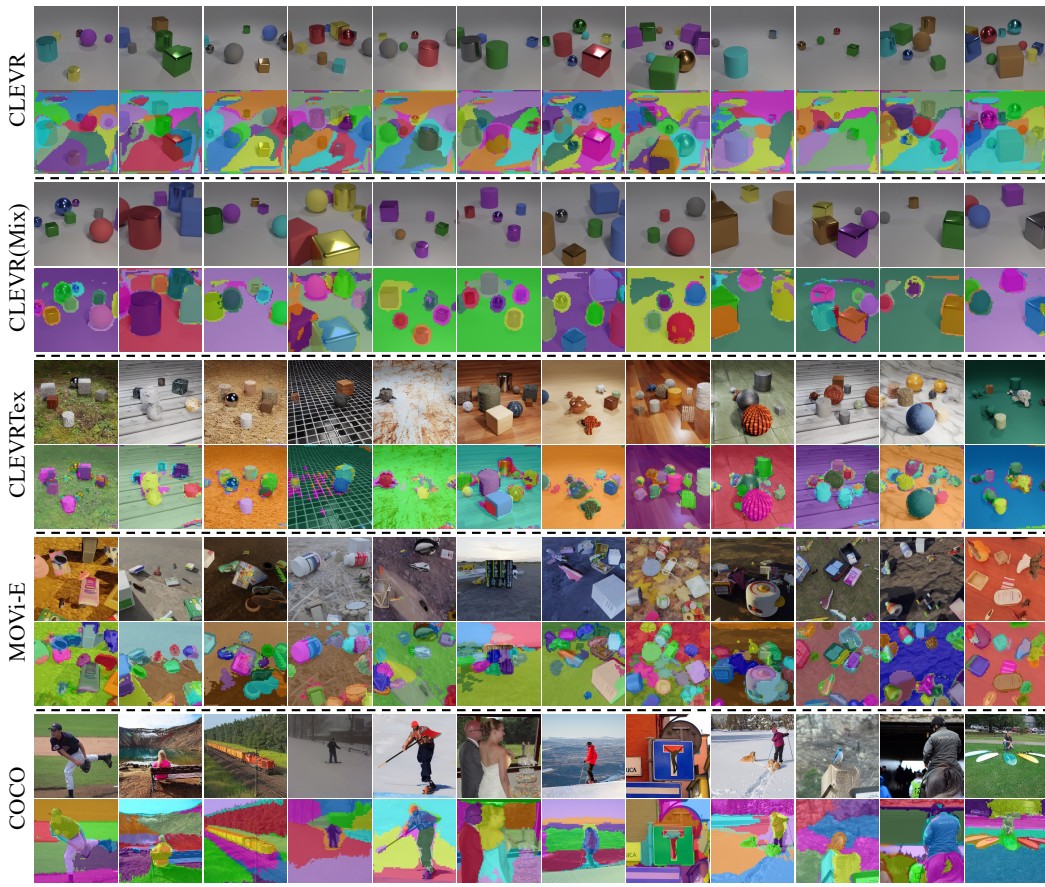

Figure 7: **Qualitative Results of Control-LSD on CLEVR, CLEVRTex, MOVi-E, and COCO.**
The CLEVR(Mix) corresponds to the model trained on both CLEVR and CLEVRTex samples.

scenarios, provide the necessary learning signal to acquire object-centric representations? In this section, we provide a preliminary exploration on this question by introducing two LSD variants: Control-LSD and Latent Slot Stable Diffusion (Stable-LSD).

### C.1 Control-LSD

**Model Design** Control-LSD utilizes a similar design of the adapter structure proposed in ControlNet [88] and consists of three main components: (1) a pre-trained Stable Diffusion model that is frozen throughout the training, (2) a "control block" that is a trainable copy of the unet encoder from Stable Diffusion, and (3) a object-centric encoder from the LSD model. Similar to LSD, the object-centric encoder takes an image as input and generates a set of slots, which are then conditioned on using a Slot-Conditioned Transformer. However, instead of providing the slots to the diffusion decoder, Control-LSD feeds the slots into the trainable control block while solely employing null text conditioning in the Stable Diffusion. To integrate the slot control signal, the outputs of each intermediate control block are element-wise added to the intermediate blocks and skip-connections within the frozen Stable Diffusion model. Following LSD, we use a input resolution of $256 \times 256$ and attention map resolution of $64 \times 64$ for all datasets. We also use the same number of slots as in LSD for multi-object dataset, and 10 slots for the COCO dataset.

**Results** We evaluate Control-LSD on three multi-object datasets: CLEVR, CLEVRTex, and MOVi-E, as well as one real-world dataset, COCO [48]. Following the proposed LSD, we also include a CLEVR(Mix) variant for the CLEVR dataset which was trained on both CLEVR and CLEVRTex samples. The segmentation results are presented in Figure 7. We can see that Control-LSD struggles to achieve satisfactory object segmentation on the CLEVR dataset. The model has difficulties in distinguishing between the background and the objects. We also note that the model demonstrate

more reasonable results on CLVERTex and introducing samples from CLEVRTex to the training process of CLEVR (i.e., CLEVR(Mix)) also led to an improvement on CLEVR, but the results are still inferior to those achieved by LSD. Interestingly, we also observe that Control-LSD achieve better segmentation quality when applied to more realistic scenes, such as those in the MOVi-E dataset. We conjecture that this is attributed to the fact that the complexity of MOVi-E aligns more closely with real-world images used for pre-training the Stable Diffusion model, as compared to the samples from CLEVR and CLEVRTex datasets. This indicates that the distribution gap between the data used for pre-training Stable Diffusion and the multi-object datasets used for learning object-centric representation might be a potential contributing factor to the lower performance of Control-LSD on the multi-object datasets.

Finally, we also evaluate Control-LSD on a real-world multi-object dataset COCO [48]. Notably, Control-LSD showcases its ability to capture semantically meaningful entities in the slots, as illustrated in the segmentation results in Figure 7. However, we find it challenging to obtain accurate object segmentations aligned with COCO annotations. In this regard, we would like to share two key observations when applying the model on real-world scenes: (1) Without human annotations, unsupervised models have difficulty distinguishing between discrete parts of an object and the entire object. This is known as the part-whole ambiguity. For instance, while COCO annotates an entire human as one segment, Control-LSD might segment it into two distinct parts: the upper body and the legs. It is important to emphasize that neither of the two segmentation results are incorrect, as each could be beneficial for specific downstream tasks. (2) The segmentation granularity is sensitive to the number of slots. This sensitivity can be attributed to the model's tendency to always utilize the entire set of slots to segment the image. Consequently, if the slot count is less than the number of entities in an image, the model tends to produce semantic masks rather than object-level segments. Conversely, if the slots outnumber the entities, the model tends to over-segment the image, i.e., splitting some of the entity into multiple slots. This behavior limits the applicability of Control-LSD in real-world data, where the true number of objects is not known to the model. Therefore, we believe further investigation is needed for applying the model to real-world applications.

## C.2   Stable-LSD

While Control-LSD bypasses the need to train a Diffusion decoder, the computational cost of training the adapter network—which includes both the control block and the object encoder—is still substantial. As an illustration, the memory demands are so significant that even with four 48GB GPUs, our experiments could only handle a training batch size of 32, which also leads to significant training latency. To address this problem, we introduce the the second LSD variant which further reduce trainable modules, we name it Latent Slot Stable Diffusion, or Stable-LSD for short. We show that despite its simple architecture, Stable-LSD demonstrate strong performance on real-world scenarios. We will start with describing the model design. [4]

**Model Design** Firstly, following the DINOSAUR model [69], we employ a pre-trained image encoder, DINOv2 [59], as the CNN backbone for the object encoder. This encoder remains frozen during training. It is important to emphasize that unlike DINOSAUR, Stable-LSD is a generative model capable of generating images. Secondly, the learned slots are provided directly to the cross attention layers of pre-trained DMs as special text tokens. This means that the slot attention module acts dual purposes — as an object encoder extracting object-centric representations and as a feature adapter converting input tokens from the image to the text domain. Notably, within the Stable-LSD framework, only the slot attention module requires training. The resulting design significantly reduces the computational demands observed with Control-LSD. In our experiment, we were able to train the model with a batch size of 128 using two 48GB GPUs.

**Real-World Segmentation Results** Despite its simplicity, Stable-LSD demonstrate its effectiveness in real-world object-centric learning. To evaluate its object-centric learning ability, we compare Stable-LSD with DINOSAUR on instance-level FG-ARI and mBO. The result can be find in Table 4. The results show that Stable-LSD achieves comparative performance to DINOSAUR with relatively superior FG-ARI value and a diminished mBO value. We show the mask visualization on Figure 8.

---

[4]The Stable-LSD implementation will be available in the official release: `https://github.com/JindongJiang/latent-slot-diffusion`

Table 4: **Quantitative Evaluation of Stable-LSD on COCO dataset.** We compare Stable-LSD with DINOSAUR [69] on instance-level FG-ARI and mBO. The results of DINOSAUR is copied from the original paper.

|  | Stable-LSD (Ours) | DINOSAUR |
|---|---|---|
| mBO (↑) | 30.38 | **31.60** |
| FG-ARI (↑) | **35.02** | 34.10 |

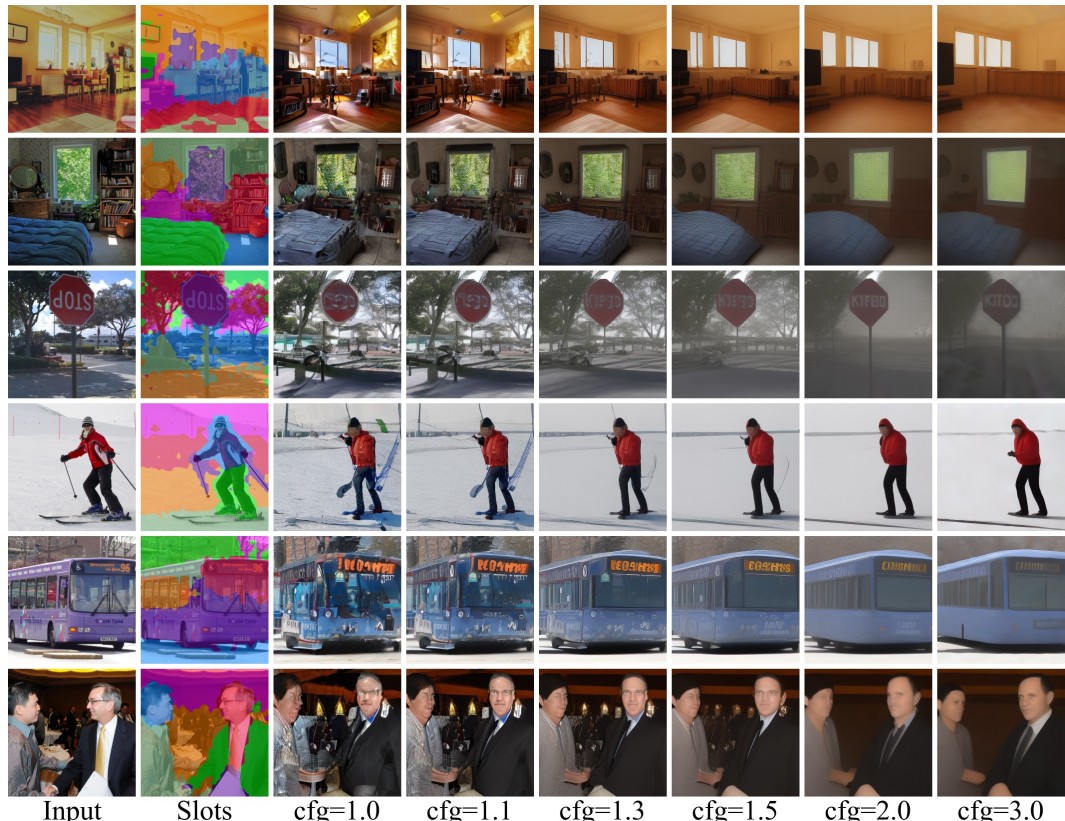

| Input | Slots | cfg=1.0 | cfg=1.1 | cfg=1.3 | cfg=1.5 | cfg=2.0 | cfg=3.0 |

Figure 8: **Unsupervised Image Segmentation and Generation Results of Stable-LSD on COCO dataset.** For image generation, we apply different cfg values on the learned slots with cfg = 1.0 equivalent to image reconstruction.

**Real-World Image Generation Results** We further evaluate the image generation quality of Stable-LSD. In Figure 8, we show the image reconstruction results (denoted by cfg = 1.0). The results demonstrate that the model output suffers from visual artifacts which constrained both the image reconstruction and generation capabilities. To further improve the generation results, we explore the technique of classifier-free guidance which is widely employed in text-to-image diffusion models.

Classifier-free guidance [34] uses a mixing weight (abbreviated as cfg) to control the weighted sum of noise predictions between a conditional and an unconditional diffusion model. With $cfg = 1$, it operates similarly to standard conditional generation. When $cfg > 1$, the model generates images that are more align to the text input, but comes with a cost of reduced image diversity. Practically speaking, an optimal $cfg > 1$ can yield images that not only better match the conditions but also exhibit enhanced quality. Since Stable-LSD provides object slots as text tokens to the pre-trained DMs, classifier-free guidance can be applied directly to the input slot, akin to how it's applied to input text in standard DMs, without any model modifications. And our experiment shows that a $cfg > 1$ for slots can also significantly improve the image generation quality.

In Figure 8, we tested various cfg values with Stable-LSD to optimize image generation quality. As we can see in the figure, from cfg = 1.0 to 1.3, the images appear cleaner with less visual artifacts,

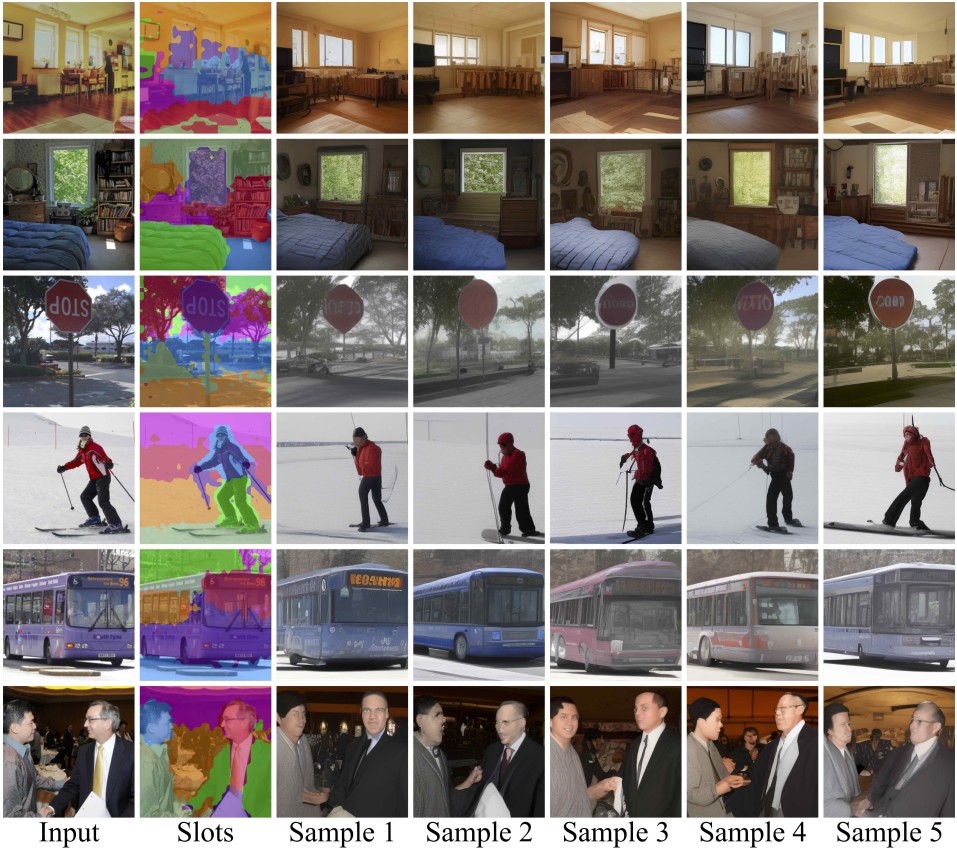

| Input | Slots | Sample 1 | Sample 2 | Sample 3 | Sample 4 | Sample 5 |

Figure 9: **Real-World Image Generations of Stable-LSD.** Each image is conditioned on the same set of learned slots and with a different initial noise map in the denoising process. We use cfg = 1.3 for all samples.

delivering conceptually more consistent images with the input slots. Nevertheless, we also observe that higher cfg values strip away intricate visual details, leading to overly simplified images. We settled on a cfg value of 1.3 for following experiments.

In Figure 9, we show image samples generated based on the same set of learned slots. The variance in these samples comes from differing the initial noise maps in the denoising process. These results demonstrate that with a proper cfg value, Stable-LSD achieves slot-based conditional image generation with unconstrained real-world objects, which is for the first time in object-centric generative models. In addition, beyond showing the diversity of the generated images, these results also underscore a crucial point from the representation learning prospective: the learned slots contain rich semantic information rather than merely performing texture-based clustering. An evidence to this is the model's ability to maintain overall semantic consistency while exhibiting diverse presentations of the same semantic content across samples. For instance, even if the background of the stop sign images shows different tree shapes, they all clearly have trees in the background. Achieving this level of consistency would be challenging if the slots lacked an abstract semantic understanding of the objects in the scene. We show additional real-world segmentation and generation results in Figures 18 - 23.

We believe the prelimilary attempts detailed in this section to be valuable explorations toward large-scale object-centric learning, and we hope our insights pave the way for further research in this area.

Table 5: **Full Table of Segmentation Performance and Representation Quality.** Extended results from the main text, now including standard deviation values.

(a) CLEVRTex

| Segmentation | SLATE | SLATE$^+$ | LSD (Ours) | Representation | SLATE | SLATE$^+$ | LSD (Ours) |
|---|---|---|---|---|---|---|---|
| mBO ($\uparrow$) | $50.88 \pm 0.70$ | $54.90 \pm 1.15$ | $\mathbf{63.93} \pm 1.93$ | Shape ($\uparrow$) | $74.24 \pm 2.41$ | $71.63 \pm 4.94$ | $\mathbf{80.23} \pm 1.59$ |
| mIoU ($\uparrow$) | $49.54 \pm 0.70$ | $52.96 \pm 1.40$ | $\mathbf{62.52} \pm 1.88$ | Material ($\uparrow$) | $69.73 \pm 2.04$ | $63.61 \pm 2.40$ | $\mathbf{75.56} \pm 1.00$ |
| FG-ARI ($\uparrow$) | $44.19 \pm 1.80$ | $\mathbf{70.71} \pm 3.35$ | $64.41 \pm 8.53$ | Position ($\downarrow$) | $1.28 \pm 0.10$ | $1.26 \pm 0.15$ | $\mathbf{1.13} \pm 0.04$ |

(b) MOVi-C

| Segmentation | SLATE | SLATE$^+$ | LSD (Ours) | Representation | SLATE | SLATE$^+$ | LSD (Ours) |
|---|---|---|---|---|---|---|---|
| mBO ($\uparrow$) | $39.37 \pm 0.80$ | $38.17 \pm 1.29$ | $\mathbf{45.57} \pm 0.80$ | Position ($\downarrow$) | $1.37 \pm 0.06$ | $1.28 \pm 0.07$ | $\mathbf{1.14} \pm 0.03$ |
| mIoU ($\uparrow$) | $37.75 \pm 0.71$ | $36.44 \pm 1.35$ | $\mathbf{44.19} \pm 0.91$ | 3D B-Box ($\downarrow$) | $1.48 \pm 0.03$ | $\mathbf{1.44} \pm 0.11$ | $\mathbf{1.44} \pm 0.02$ |
| FG-ARI ($\uparrow$) | $49.54 \pm 1.44$ | $\mathbf{52.04} \pm 0.45$ | $51.98 \pm 3.53$ | Category ($\uparrow$) | $42.45 \pm 0.13$ | $45.32 \pm 1.13$ | $\mathbf{46.11} \pm 0.59$ |

(c) MOVi-E

| Segmentation | SLATE | SLATE$^+$ | LSD (Ours) | Representation | SLATE | SLATE$^+$ | LSD (Ours) |
|---|---|---|---|---|---|---|---|
| mBO ($\uparrow$) | $30.17 \pm 1.70$ | $22.17 \pm 0.38$ | $\mathbf{38.96} \pm 0.47$ | Position ($\downarrow$) | $2.09 \pm 0.13$ | $2.15 \pm 0.07$ | $\mathbf{1.85} \pm 0.05$ |
| mIoU ($\uparrow$) | $28.59 \pm 1.66$ | $20.63 \pm 0.35$ | $\mathbf{37.64} \pm 0.45$ | 3D B-Box ($\downarrow$) | $3.36 \pm 0.12$ | $3.37 \pm 0.27$ | $\mathbf{2.94} \pm 0.003$ |
| FG-ARI ($\uparrow$) | $46.06 \pm 3.32$ | $45.25 \pm 0.77$ | $\mathbf{52.17} \pm 0.89$ | Category ($\uparrow$) | $38.93 \pm 0.16$ | $38.00 \pm 0.36$ | $\mathbf{42.96} \pm 0.21$ |

## D  Additional Implementation Details

To ensure reproducibility, we are sharing additional implementation details of LSD in this section. Our implementation will be made available at `https://github.com/JindongJiang/latent-slot-diffusion`

### D.1  Experiment Setup

**Datasets.** In our experiments, we create three data subsets for each dataset: training, validation, and testing. For CLEVR, we utilize the official split for training and validation sets. However, for evaluation, we do not use the official test set as CLEVR does not provide ground-truth object segmentation masks. Instead, we use the "clevr_with_mask" subset from the Multi-Object Datasets [38] for evaluation. This subset is designed to have the same object distribution as CLEVR while providing ground-truth masks and attributes annotations for each object in the image, which allows us to evaluate the segmentation and representation quality of each model. We use the full subset as the test set for evaluation. For CLEVRTex, there is no official split provided, so we allocate 80% of the data for training, 10% for validation, and 10% for testing. Regarding MOVi-C and MOVi-E, we use 90% of the training set data for training and reserve 10% for validation. We use the official validation set for testing. It is important to note that the test sets in MOVi-C and MOVi-E are designed for out-of-distribution (OOD) evaluation, so we do not use them for test time evaluation. For the FFHQ dataset, we use 86% of the dataset ($\sim$ 60K images) for training and 7% ($\sim$ 5K images) for validation. To ensure consistency across all models, we shuffled the dataset and utilized a fixed random seed when splitting the data.

**Segmentation Metrics.** To evaluate the unsupervised object segmentation, we use three metrics: the foreground adjusted rand index (FG-ARI), the mean intersection over union (mIoU), and the mean best overlap (mBO). We predict the object mask by using the argmax along the slot dimension in the attention map. In computing the mIoU, we use Hungarian matching to obtain the ground-truth and slots assignment. Following [15], we use the negative cosine similarity as the matching loss. In mBO, we assign the ground-truth segment to the slot with the largest overlap. Note that the mBO metric is generally simpler than mIoU since it does not require a strict one-to-one mapping between the ground-truth and predicted masks.

### D.2  Implementation Details

We will provide an overview of the implementation details in this section. The hyperparameters used in our approach are listed in Table 6.

**Auto-Encoding CNN backbone.** The object-centric encoder in our model includes a CNN backbone, which transforms the raw image input into feature vectors that can be processed by the slot attention

| Module | Hyperparameter | CLEVR | CLEVRTex | MOVi-C/E | FFHQ |
|---|---|---|---|---|---|
| | | Dataset | | | |
| General | Batch Size | 64 | 64 | 128 | 128 |
| | Training Steps | 200K | 200K | 200K | 200K |
| | # K-means Clusters | 5 | 5 | 18 | 18 |
| CNN Backbone | Input Resolution | 256 | | | |
| | Output Resolution | 64 | | | |
| | Self Attention | Middle Layer | | | |
| | Base Channels | 128 | | | |
| | Channel Multipliers | [1,1,2,4] | | | |
| | # Heads | 8 | | | |
| | # Res Blocks / Layers | 2 | | | |
| | Output Channels | 128 | 128 | 192 | 192 |
| | Learning Rate | 0.0001 | 0.00003 | 0.0001 | 0.0001 |
| Slot Attention | Input Resolution | 64 | | | |
| | # Iterations | 3 | | | |
| | Slot Size | 128 | 128 | 192 | 192 |
| | # Slots | 11 | 11 | 24 | 4 |
| | Learning Rate | 0.0001 | 0.00003 | 0.0001 | 0.0001 |
| Auto-Encoder | Model | KL-8 | | | |
| | Input Resolution | 256 | | | |
| | Output Resolution | 32 | | | |
| | Output Channels | 4 | | | |
| LSD Decoder | Input Resolution | 32 | | | |
| | Input Channels | 4 | | | |
| | $\beta$ scheduler | Linear | | | |
| | Mid Layer Attention | Yes | | | |
| | # Res Blocks / Layers | 2 | | | |
| | Image Latent Scaling | 0.18215 | | | |
| | Learning Rate | 0.0001 | | | |
| | # Heads | 4 | 4 | 8 | 8 |
| | Base Channels | 128 | 128 | 192 | 192 |
| | Attention Resolution | [1,2,4] | [1,2,4] | [1,2,4,4] | [1,2,4,8] |
| | Channel Multipliers | [1,2,4] | [1,2,4] | [1,2,4,4] | [1,2,4,4] |

Table 6: Hyperparameters of our model used in our experiments.

module. Previous studies have used multi-layer CNN networks as image encoders, as demonstrated by [53, 70, 73]. However, in our experiments with Latent Diffusion Models, we have found that such encoders tend to produce low-level features that mainly represent pixel position or color information. This leads to the slot attention module grouping pixels based on these low-level features, rather than on high-level object information. We hypothesize that this problem may arise from the early stages of training, where the model lacks a semantic understanding of the input image. This results in the LSD Decoder relying only on low-level and local information from slots to denoise the noisy input, without encouraging the object-centric encoder to learn semantic-level pixel grouping. As a result, without additional guidance, the model may become trapped in a suboptimal solution. We have observed this issue only when a Diffusion Decoder is used.

To overcome this issue, we have incorporated a UNet architecture [64] as the image encoder in the object-centric encoder. The main advantage of this approach is that the auto-encoding structure of UNet allows high-level global context information to be blended into the output features and slots, facilitating the learning process and helping the model escape from suboptimal pixel groupings. Additionally, the skip connections in UNet allow the output features to retain fine-grained low-level details from early CNN layers, ensuring that the resulting representations are rich in both high-level object information and low-level texture information. These improvements to the object-centric

encoder result in more accurate object-centric representations and enable better image generation results. We provide detailed design information in Table 6. Prior to applying the UNet structure, we first employ a single CNN layer with a kernel size of 4 and stride of 4 to downsample the image from $256 \times 256 \times 3$ to $64 \times 64 \times S$, where $S$ is the base channel size in UNet. In Appendix E.1, we provide an additional ablation study for the UNet encoder.

**Slot-Conditioned Cross Attention.** We implement the conditioning mechanism between the object slots and decoder features through a transformer network as in LDM [63] and SLATE [70]. This conditioning mechanism consists of two transformer layers, including a self-attention layer that computes self-attention within the features generated by the UNet in the LSD Decoder, followed by a cross-attention layer that computes cross-attention between the object slots and UNet features.

In the cross-attention layer, we use the object slots $\mathbf{S}$ as key-value pairs, and the queries are intermediate layers of the diffusion network augmented with positional embedding $(\tilde{\mathbf{h}}_l + \mathbf{p}_l)$. We then apply multi-head cross attention for each head as $\text{Attention}(\mathbf{Q}, \mathbf{K}, \mathbf{V}) = \texttt{softmax}\left(\frac{\mathbf{Q}\mathbf{K}^T}{\sqrt{d}}\right) \cdot V$, where

$$\mathbf{Q} = \mathbf{W}_Q^{(i)} \cdot (\tilde{\mathbf{h}}_l + \mathbf{p}_l), \ \mathbf{K} = \mathbf{W}_K^{(i)} \cdot \mathbf{S}, \ \mathbf{V} = \mathbf{W}_V^{(i)} \cdot \mathbf{S}.$$

Here, $i$ is the head index, $\tilde{\mathbf{h}}_l$ represents the intermediate layers of the diffusion network, $\mathbf{p}_l$ is the positional embedding, $\mathbf{S}$ is the set of object slots computed from the object-centric encoder, and $\mathbf{W}_V^{(i)}, \mathbf{W}_Q^{(i)}, \mathbf{W}_K^{(i)}$ are learnable projection matrices.

**Additional Details for Object-Centric Encoder.** For all multi-object datasets, we set the number of slots to be the maximum number of objects per image plus one (intended for background). For the FFHQ dataset, we segment the image into 4 slots. We use the $256 \times 256$ resolution images for all experiments and, if necessary, we use bilinear interpolation for re-scaling and center cropping to get square images. We do not introduce additional data augmentations during pre-processing. When computing the object segmentation, we use the attention map from the slot attention as the mask prediction. The resolution of the mask is designed to be $64 \times 64$ for all models.

**Additional Details for LSD Decoder.** We add learnable positional embeddings into the self and cross-attention layers of UNet, which are found to greatly improve the stability of the training process for the CLEVR and CLEVRTex datasets. However, we have observed that this modification has a minor impact on the MOVi-C/E and FFHQ datasets, but have applied it across all models to maintain consistency. For the image auto-encoder, we use the KL-8 version provided in the LDM repository. During the generation processes for all tasks, we employ the DDIM sampler with $\eta = 1$ and run it for 200 steps.

**Baseline Implementation.** Our baseline implementations are closely based on the official releases. For the SLATE baseline, we adopt the improved version proposed in [73] [5]. We empirically find this version to be more robust to complex scenes in terms of both object segmentation and generation quality. The key difference of this version from the original SLATE is in the slot encoder. The original SLATE compute object slots from the latent space of the dVAE. Therefore, the attention resolution and information granularity of the slots are constrained by the dVAE latents. The improved SLATE addresses this issue by using a jointly trained CNN backbone to directly learn the slots from the original image. As shown in [73], this improved version of SLATE achieves improved performance on complex datasets such as MOVi-C and MOVi-E. For SLATE$^+$, we replace the jointly trained dVAE with a VQGAN model pre-trained on OpenImages [6].

# E   Additional Experiments

## E.1   Ablation on CNN backbone in Object-Centric Encoder

This section presents an ablation study on the CNN backbone used in the object-centric encoder. To investigate the impact of the CNN backbone choice, we introduce a variant of LSD named LSD-CNN. Specifically, LSD-CNN uses a multi-layer CNN network as the image encoder in the object-centric encoder, following the same CNN design as in the improved SLATE model [73]. We compare the

---

[5]`https://github.com/singhgautam/steve`
[6]We use the pre-trained weights from https://ommer-lab.com/files/latent-diffusion/vq-f8.zip

Table 7: **Ablation Study on CNN backbone.** This table presents the results of an ablation study on the CNN backbone in the object-centric encoder. Specifically, we compare the performance of the LSD method that uses a UNet backbone to the performance of LSD-CNN, which utilizes a multi-layer Convnet backbone.

(a) CLEVRTex

| Segmentation | LSD-CNN | LSD | Representation | LSD-CNN | LSD |
|---|---|---|---|---|---|
| mBO (↑) | 55.61 | **66.56** | Shape (↑) | 75.00 | **81.60** |
| mIoU (↑) | 53.82 | **65.02** | Material (↑) | **84.43** | 77.77 |
| FG-ARI (↑) | 43.88 | **61.74** | Position (↓) | 1.50 | **1.10** |

(b) MOVi-C

| Segmentation | LSD-CNN | LSD | Representation | LSD-CNN | LSD |
|---|---|---|---|---|---|
| mBO (↑) | 27.43 | **46.29** | Position (↓) | 1.25 | **1.14** |
| mIoU (↑) | 26.31 | **44.99** | 3D B-Box (↓) | 1.46 | **1.44** |
| FG-ARI (↑) | 42.83 | **50.53** | Category (↑) | 41.29 | **46.71** |

(c) MOVi-E

| Segmentation | LSD-CNN | LSD | Representation | LSD-CNN | LSD |
|---|---|---|---|---|---|
| mBO (↑) | 22.00 | **39.63** | Position (↓) | 2.01 | **1.92** |
| mIoU (↑) | 20.71 | **38.28** | 3D B-Box (↓) | 3.33 | **2.94** |
| FG-ARI (↑) | 42.14 | **53.40** | Category (↑) | 35.82 | **43.15** |

results of this variant with the proposed LSD, which utilizes the UNet structure as the image encoder in the object-centric encoder, on CLEVRTex, MOVi-C, and MOVi-E. The results are presented in Table 7. Overall, we observe that LSD outperforms LSD-CNN across nearly all tasks and datasets, suggesting that the UNet CNN backbone is critical for achieving the high performance of our model.

To further clarify the architecture choice, we also explore a LSD variant named LSD-Latent that employs an image latent encoder as its CNN backbone. We evaluate LSD-Latent on MOVi-E dataset and compare its object segmentation quality with LSD. The results can be seen in Table 8. The sub-optimal performance of LSD-Latent underscores the importance of having a separate CNN encoder in the object encoder.

# F Additional Image Samples

We include additional generation samples in Figures 10 - 23. The additional samples demonstrate that LSD achieves more accurate object segmentation and generates more detailed and coherent scenes compared to the baselines. We provide additional observations and insights in the following sections.

## F.1 Visualization of Visual Concept Prompts

In this section, we present visualizations of the visual concept prompts employed for generating image samples in the CLEVRTex and FFHQ datasets, as illustrated in Figure 10. To generate these visualizations, we apply the attention map corresponding to each specific slot as a mask over the original image, which highlights the regions captured by that slot. We observe that the LSD model

Table 8: **Ablation Study on CNN backbone using Diffusion Latent Encoder.** We compare LSD with LSD-Latent on segmentation quality in the MOVi-E dataset.

| Segmentation | LSD-Latent | LSD |
|---|---|---|
| mBO (↑) | 5.52 | **39.63** |
| mIoU (↑) | 4.04 | **38.28** |
| FG-ARI (↑) | 10.11 | **53.40** |

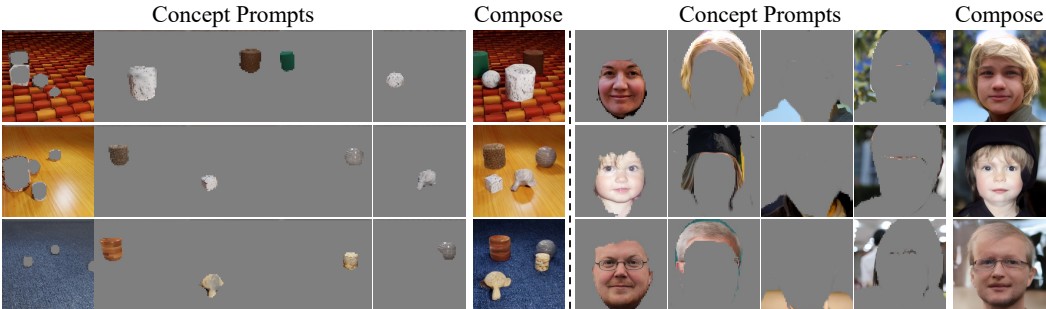

Figure 10: **Compositional Image Generation with Concept Prompts.** In this visualization, we show a concept prompt constructed by composing arbitrary slots from our visual concept library and the corresponding generated image by LSD.

can seamlessly combine components from distinct images to create a coherent image sample while preserving the attributes of each component in the final image. Note that each concept library is associated with one k-means cluster within the extracted slots, and each prompt displayed in Figure 10 is sampled from one such library. The visualization of the prompts suggests that the concept libraries is able to capture practical concept categories, such as objects and backgrounds in the CLEVRTex dataset and facial attributes, hairstyles, clothing, and backgrounds in the FFHQ dataset.

## F.2    Impact of Pre-Trained Auto-Encoder on Generation Quality

We investigate the effect of the pre-trained image auto-encoder in the comparison between SLATE and SLATE$^+$ in Figure 12 and 13. Our results show that the use of the pre-trained auto-encoder leads to less blurry images and significantly more details, such as the floor and object texture in CLEVRTex and face detail in FFHQ. However, we also observe that applying the pre-trained auto-encoder does not necessarily improve the coherence of the generated scenes. For instance, in some SLATE$^+$ samples, the shape of the objects appear distorted in CLEVRTex and the background and hair style of FFHQ images are rendered as color patches. These findings indicate the limitation of the transformer decoders in SLATE and SLATE$^+$ models and highlights the crucial role of LSD's diffusion decoder in achieving optimal generation quality.

## F.3    Impact of Segmentation Quality on Slot-Based Editing

In Figure 14 and 15, we investigate the effect of object segmentation on the slot-based editing task. Our result show that when the object segmentation is almost perfect, editing tasks such as object removing can be easily accomplished by discarding the corresponding slot during image decoding. It is important to note that the object segmentation masks are derived from the attention weights on image features. Therefore, a perfect segmentation mask indicates that the object information is completely assigned to a single slot. However, we also observe cases when the object assignments are incomplete, such as when one object is divided into multiple slots. In such cases, removing an object would require removing all associated slots. These observations emphasize the importance of accurate object segmentation in achieving effective slot-based editing. Although LSD has demonstrated superior performance compared to existing approaches, further exploration in object segmentation is necessary to extend it to real-world applications.

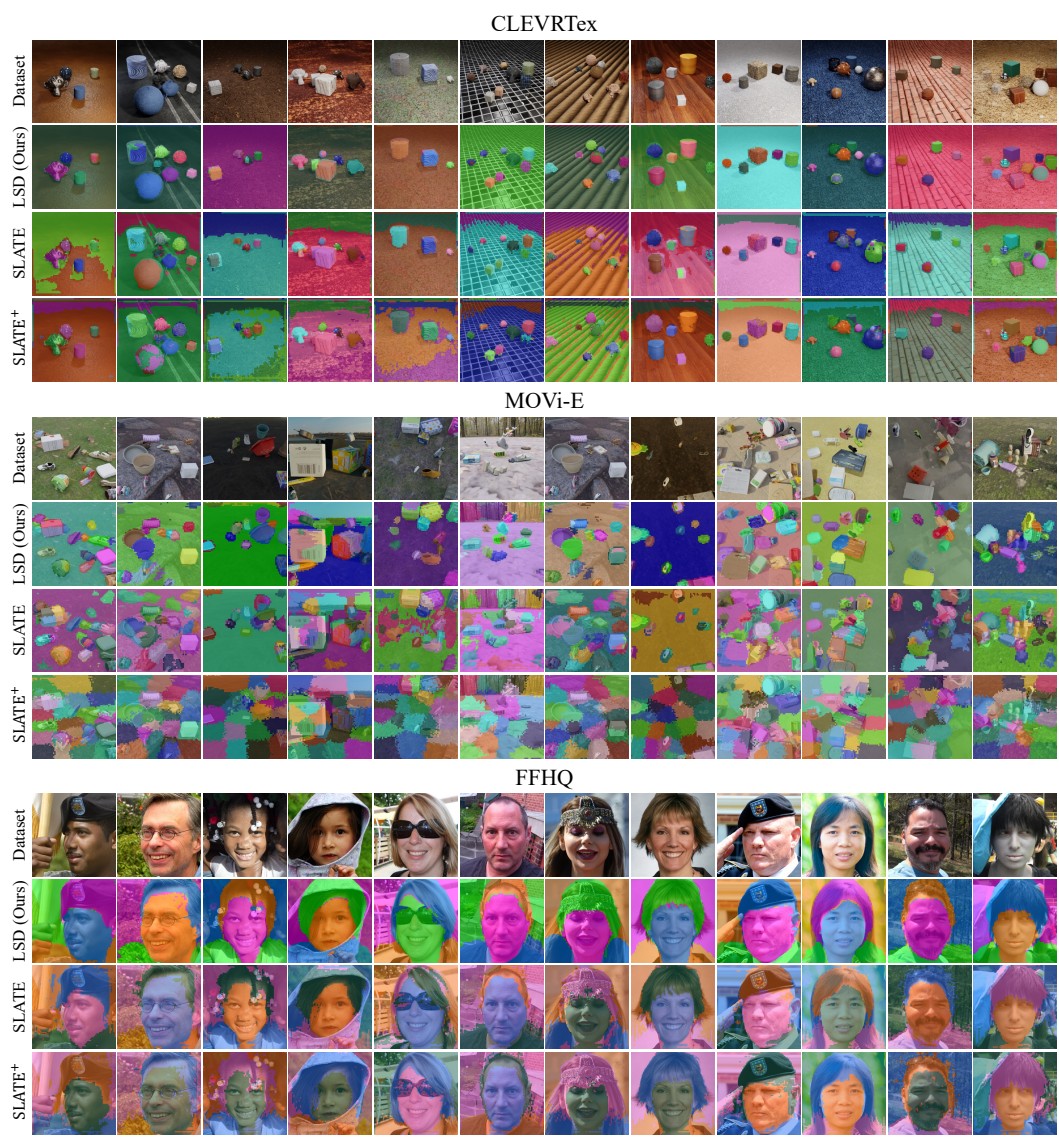

Figure 11: **Visualization of Unsupervised Object Segmentation.** LSD achieves more accuracy object segmentation when comparing to the other methods.

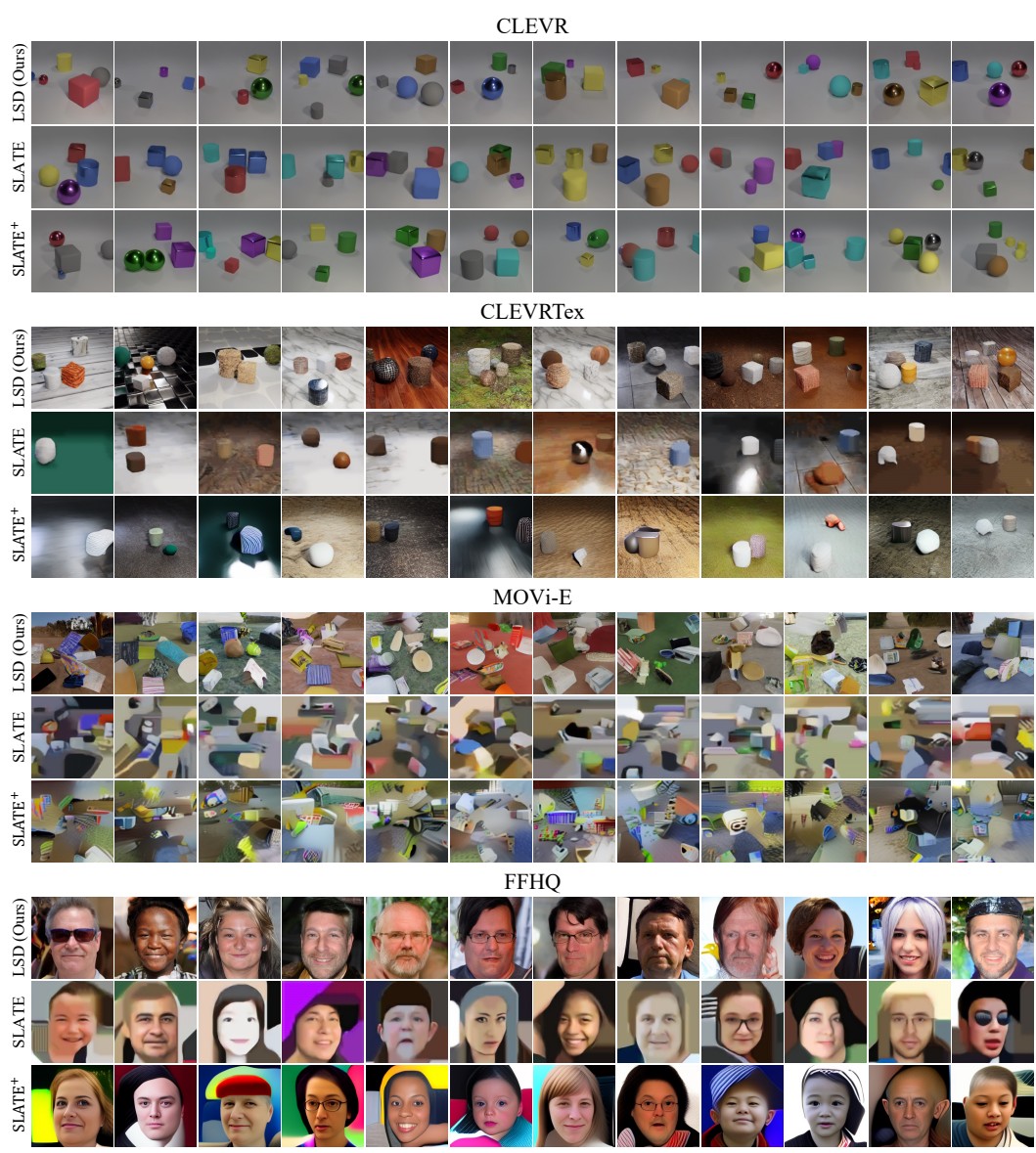

Figure 12: **Compositional Generation Samples using Visual Concept Library.** LSD provides significantly higher fidelity and more clear details when comparing to the other methods.

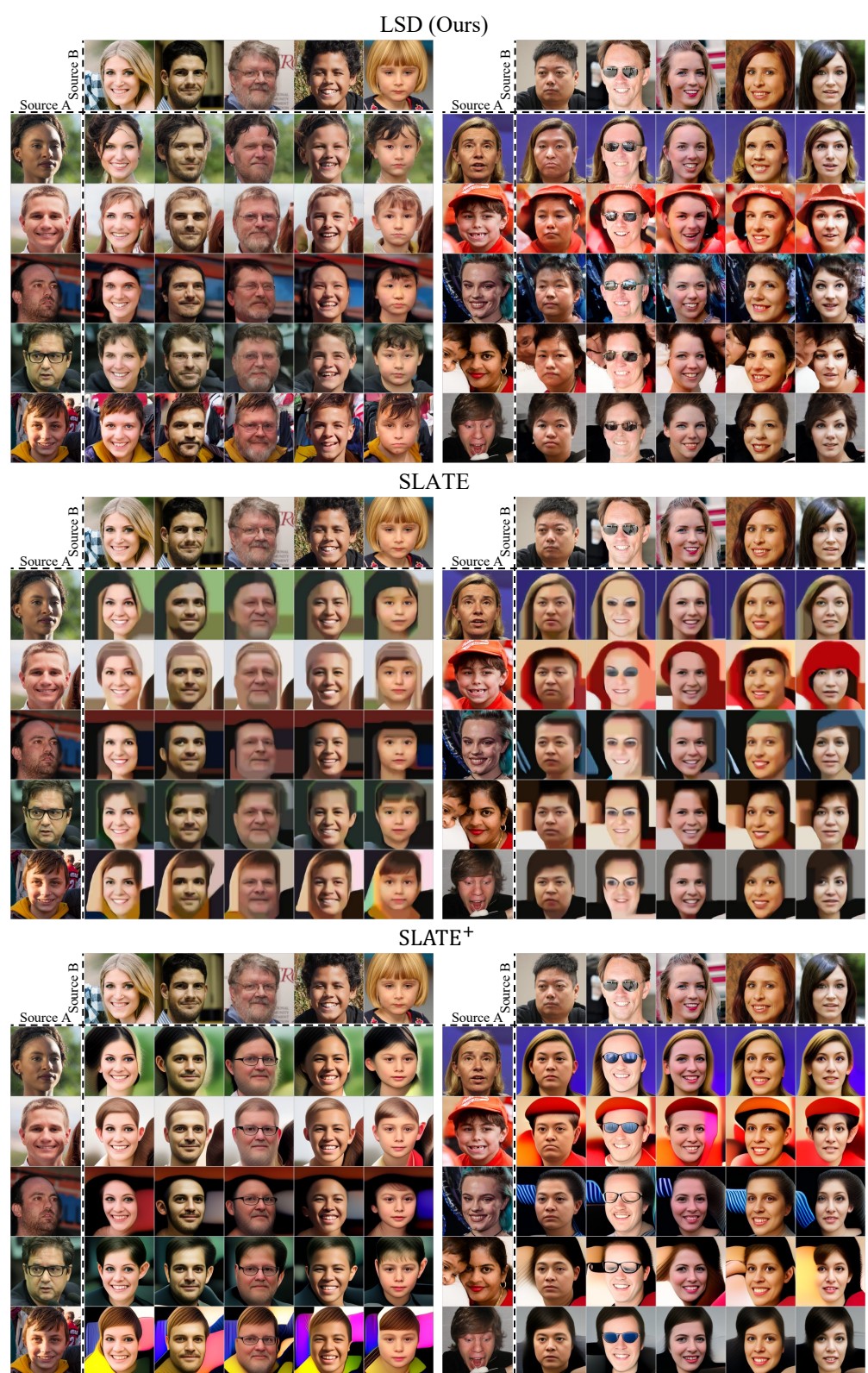

Figure 13: **Slot-Based Image Editing: Face Replacement.** We show face replacement in the FFHQ dataset, where we compose new images by combining the face slots from Source-B images with the hairstyle, clothing, and background slots from Source-A images.

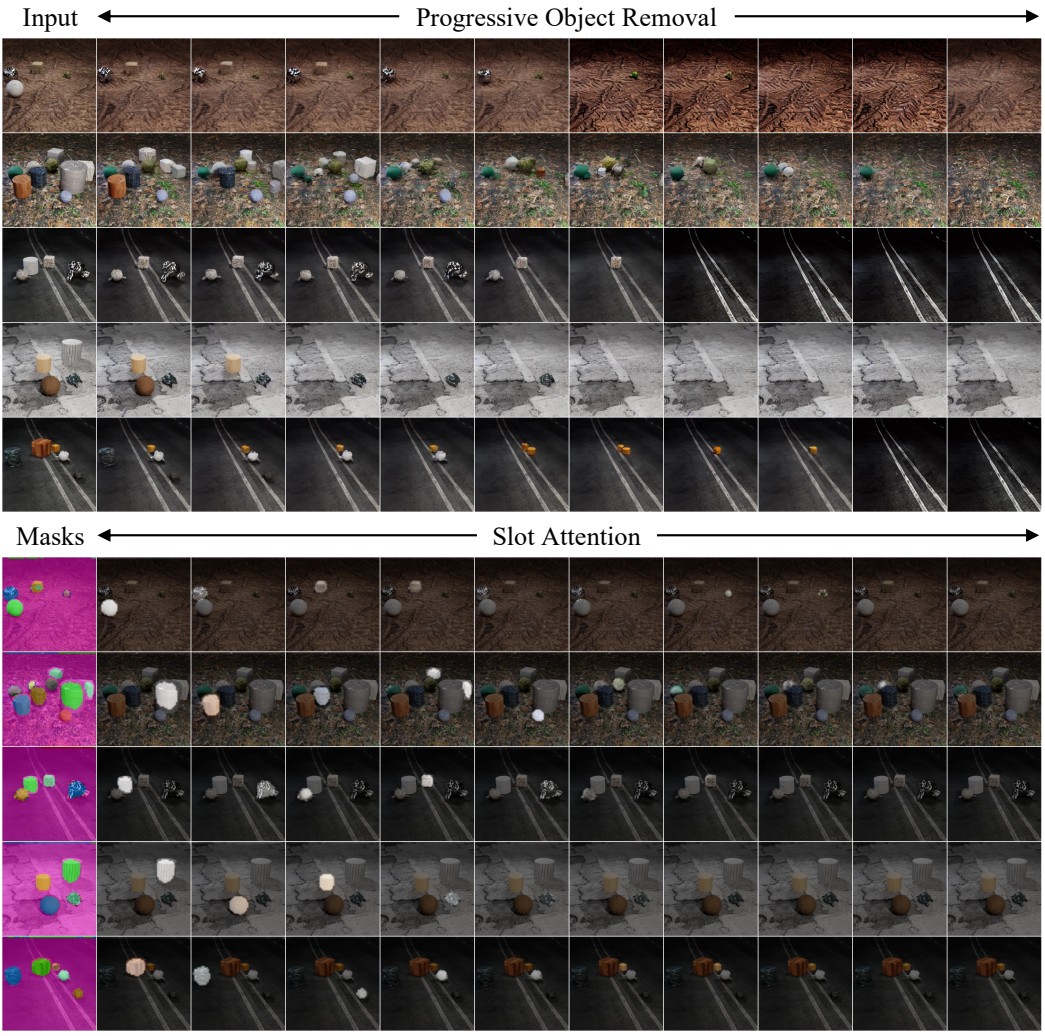

Figure 14: **Slot-Based Image Editing: Object Removal and Segmentation Mask. Sample 1.** *Top:* We show object removal by discarding the corresponding slots. *Bottom:* We show the segmentation masks of each slot.

Input ←——————————— Progressive Object Removal ——————————→

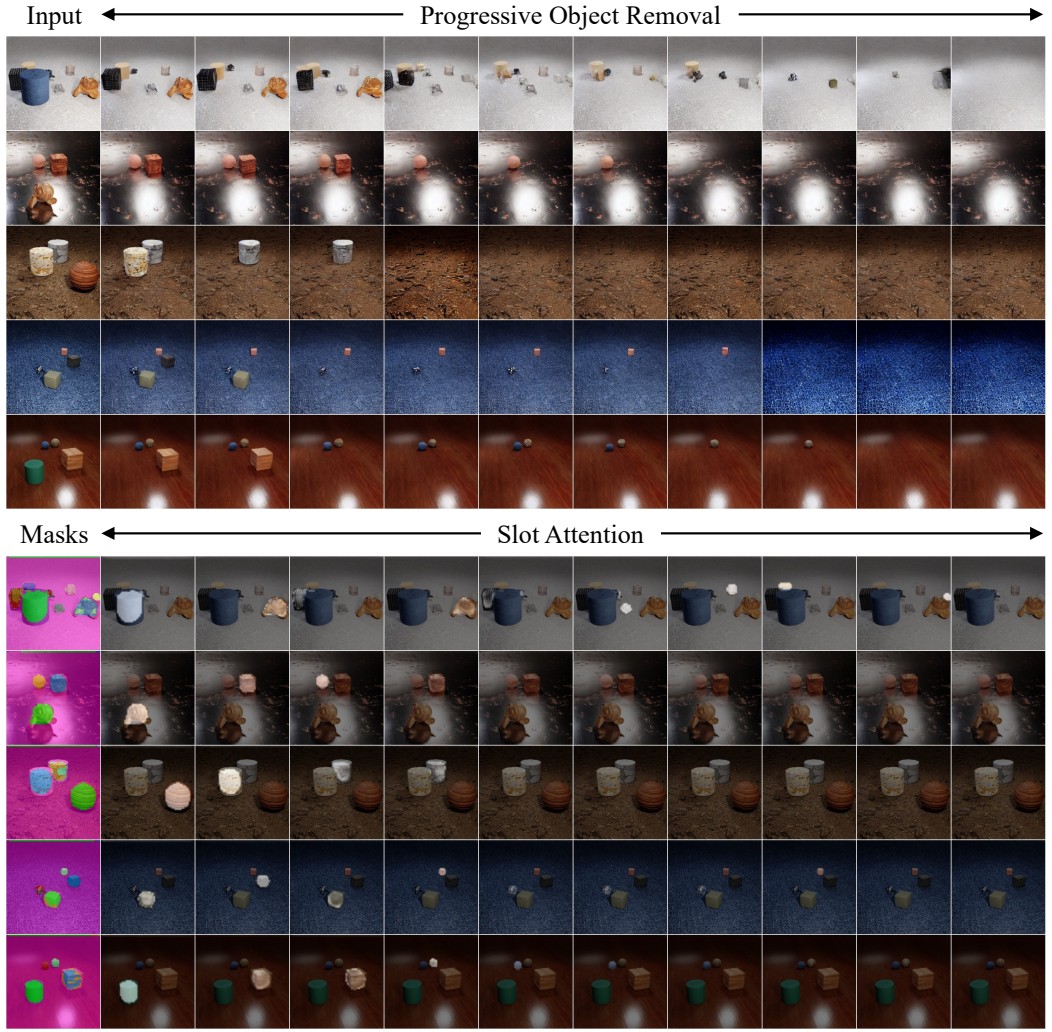

Masks ←——————————— Slot Attention ——————————→

Figure 15: **Slot-Based Image Editing: Object Removal and Segmentation Mask. Sample 2.**

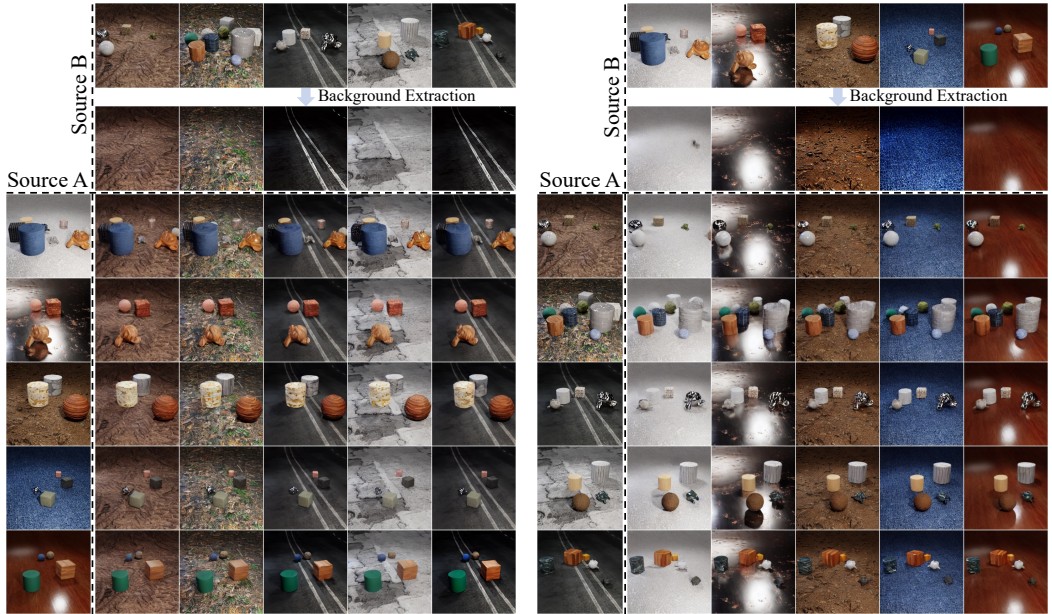

Figure 16: **Slot-Based Image Editing: Background Replacement.** The background replacement task is performed by replacing the background-associated slots.

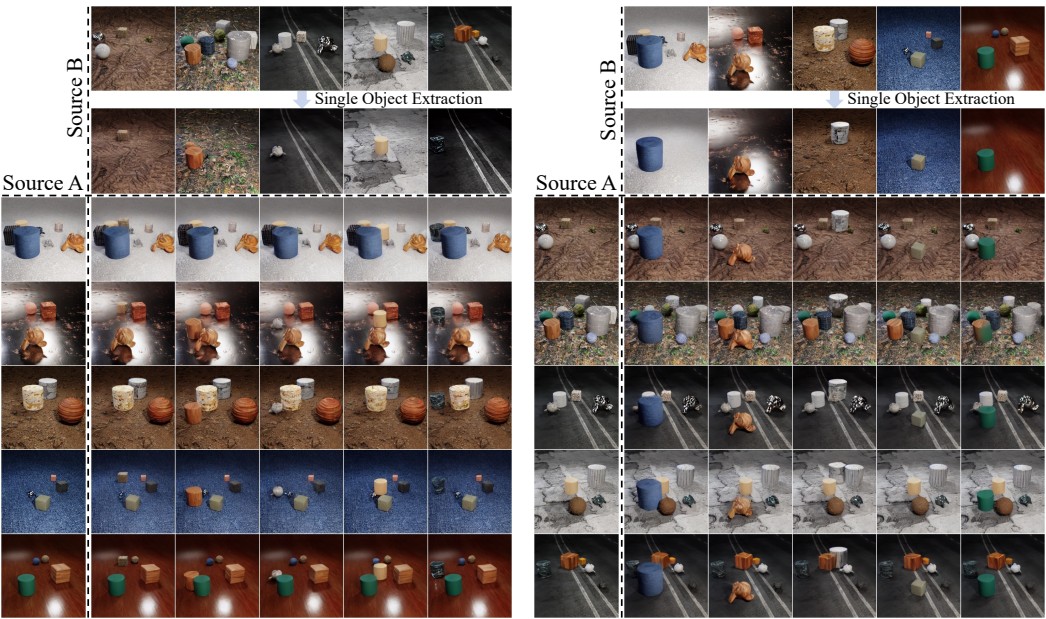

Figure 17: **Slot-Based Image Editing: Object Insertion.** The background replacement task is performed by simply adding the new slot to the existing set of slots.

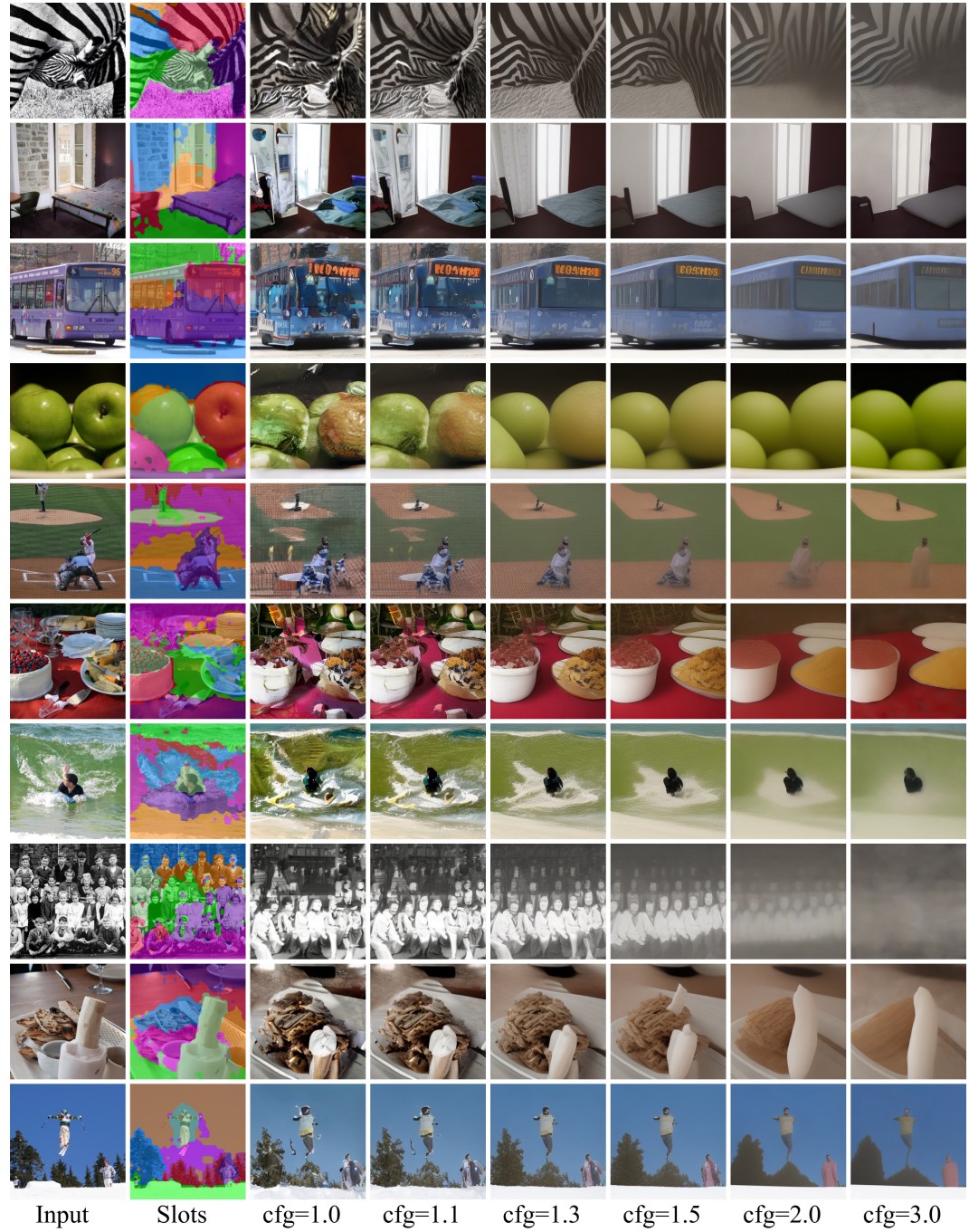

Input    Slots    cfg=1.0    cfg=1.1    cfg=1.3    cfg=1.5    cfg=2.0    cfg=3.0

Figure 18: **Random Samples for Real-World Object-Centric Learning and Conditional Generation 1.**

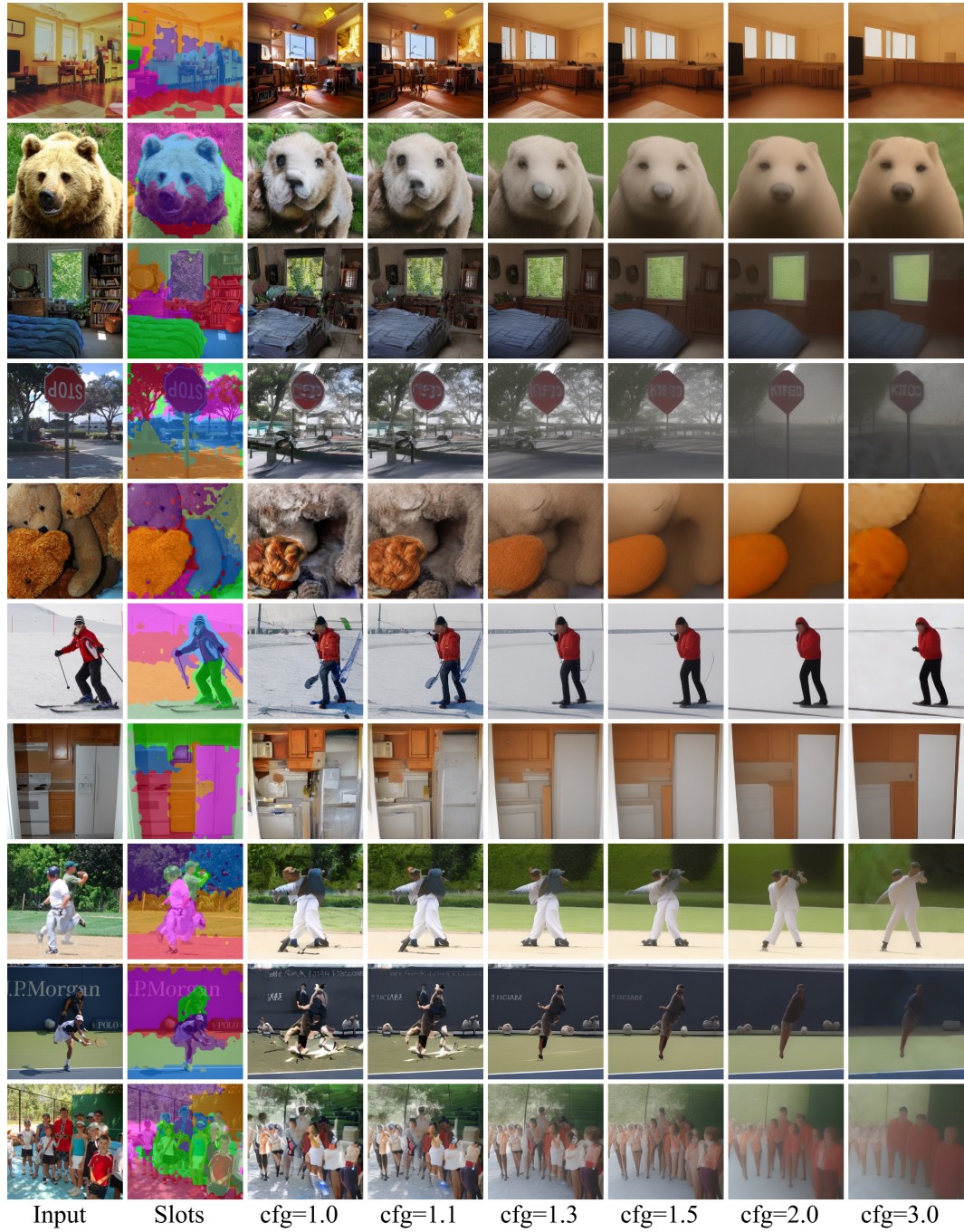

| Input | Slots | cfg=1.0 | cfg=1.1 | cfg=1.3 | cfg=1.5 | cfg=2.0 | cfg=3.0 |

Figure 19: **Random Samples for Real-World Object-Centric Learning and Conditional Generation 2.**

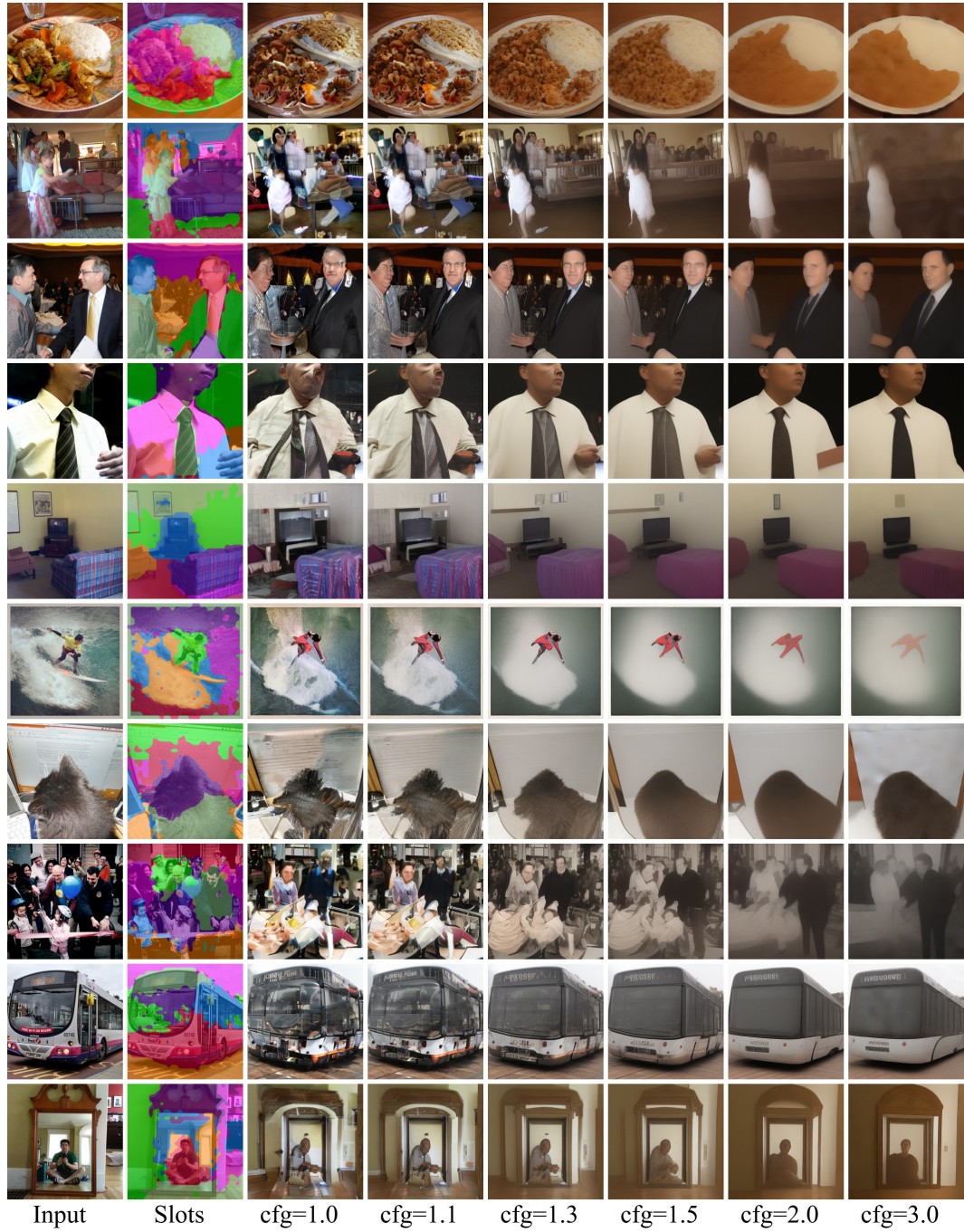

| Input | Slots | cfg=1.0 | cfg=1.1 | cfg=1.3 | cfg=1.5 | cfg=2.0 | cfg=3.0 |

Figure 20: **Random Samples for Real-World Object-Centric Learning and Conditional Generation 3.**

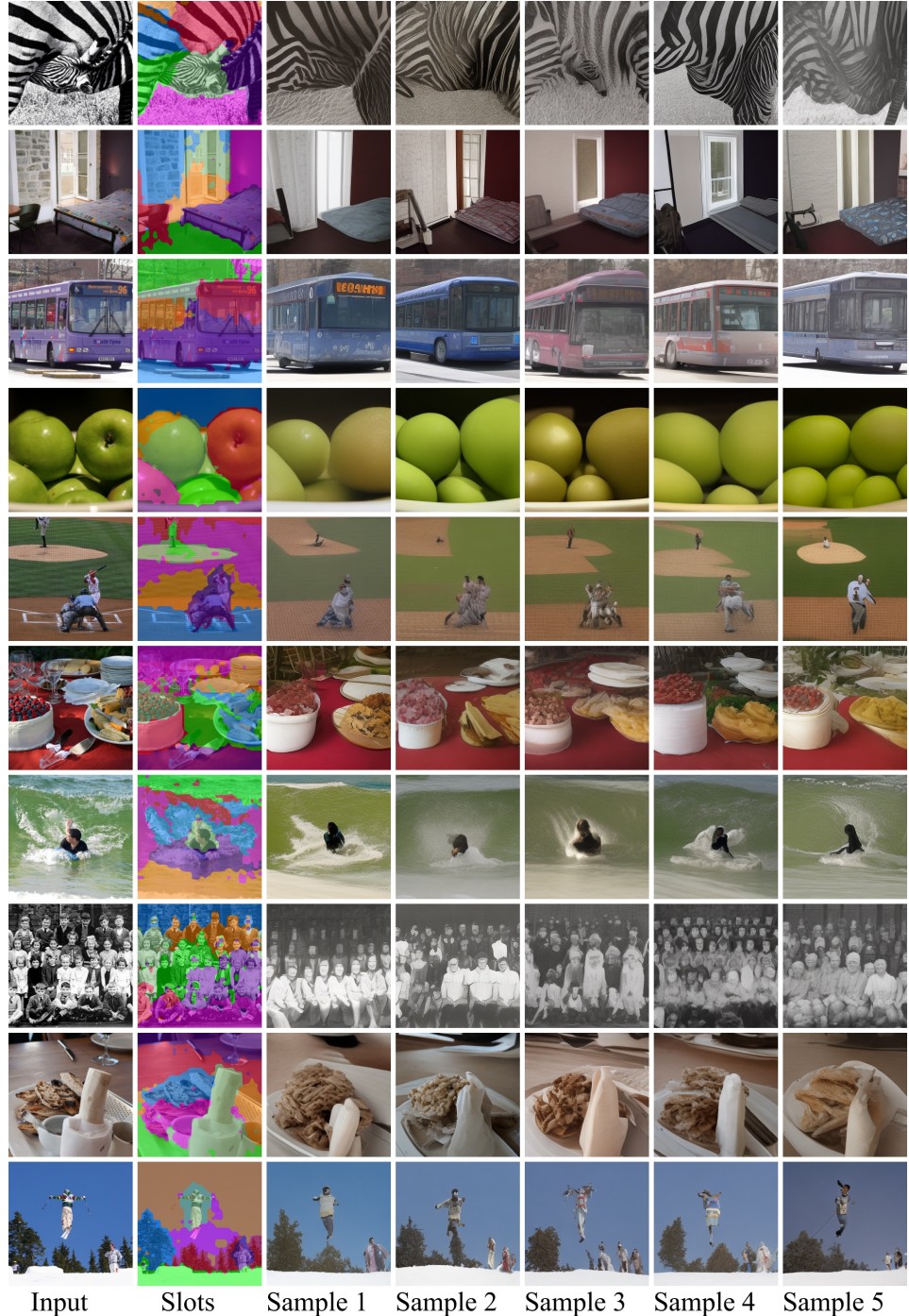

Input  Slots  Sample 1  Sample 2  Sample 3  Sample 4  Sample 5

Figure 21: **Random Samples for Real-World Image Generation 1.**

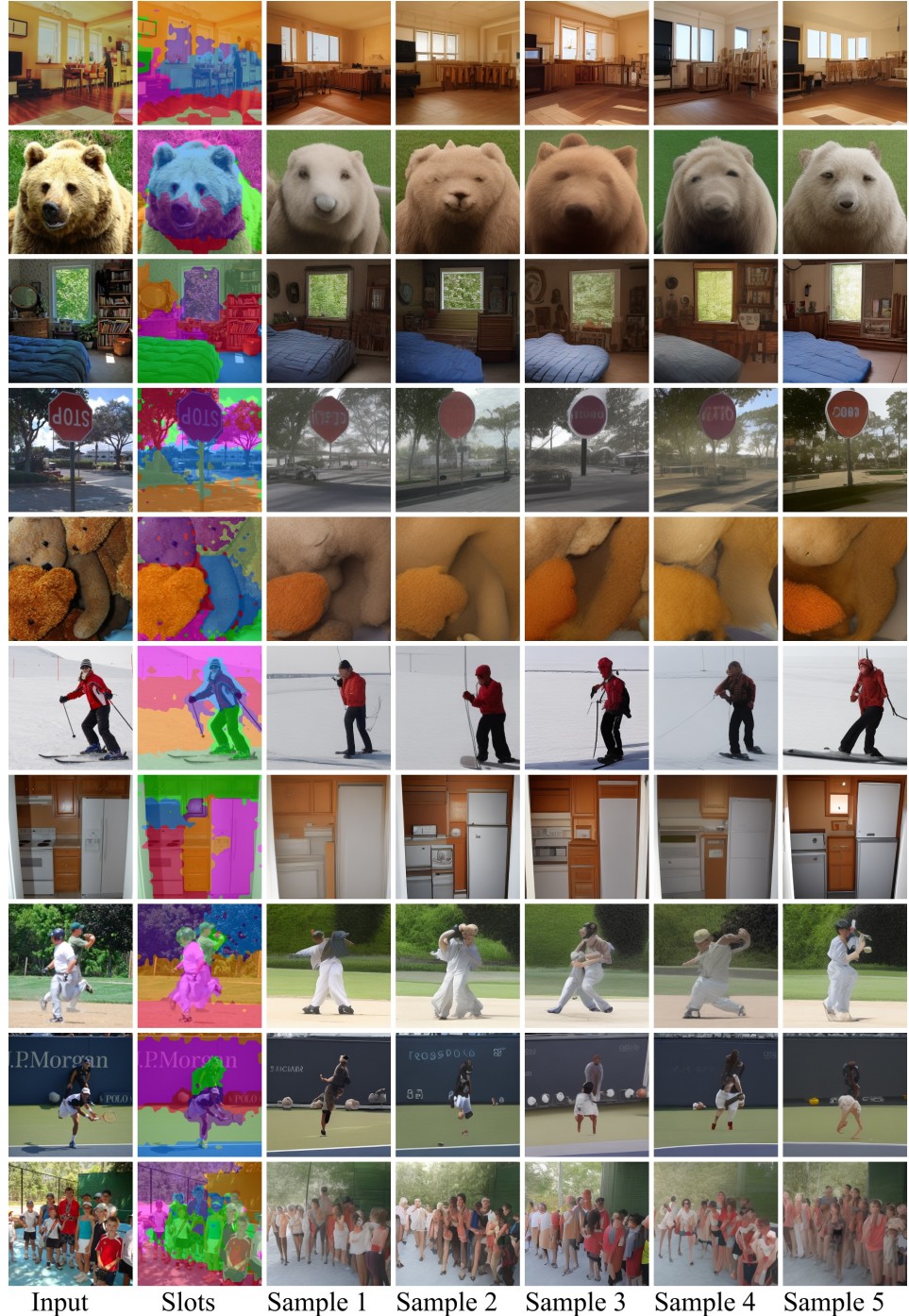

Input    Slots    Sample 1    Sample 2    Sample 3    Sample 4    Sample 5

Figure 22: **Random Samples for Real-World Image Generation 2.**

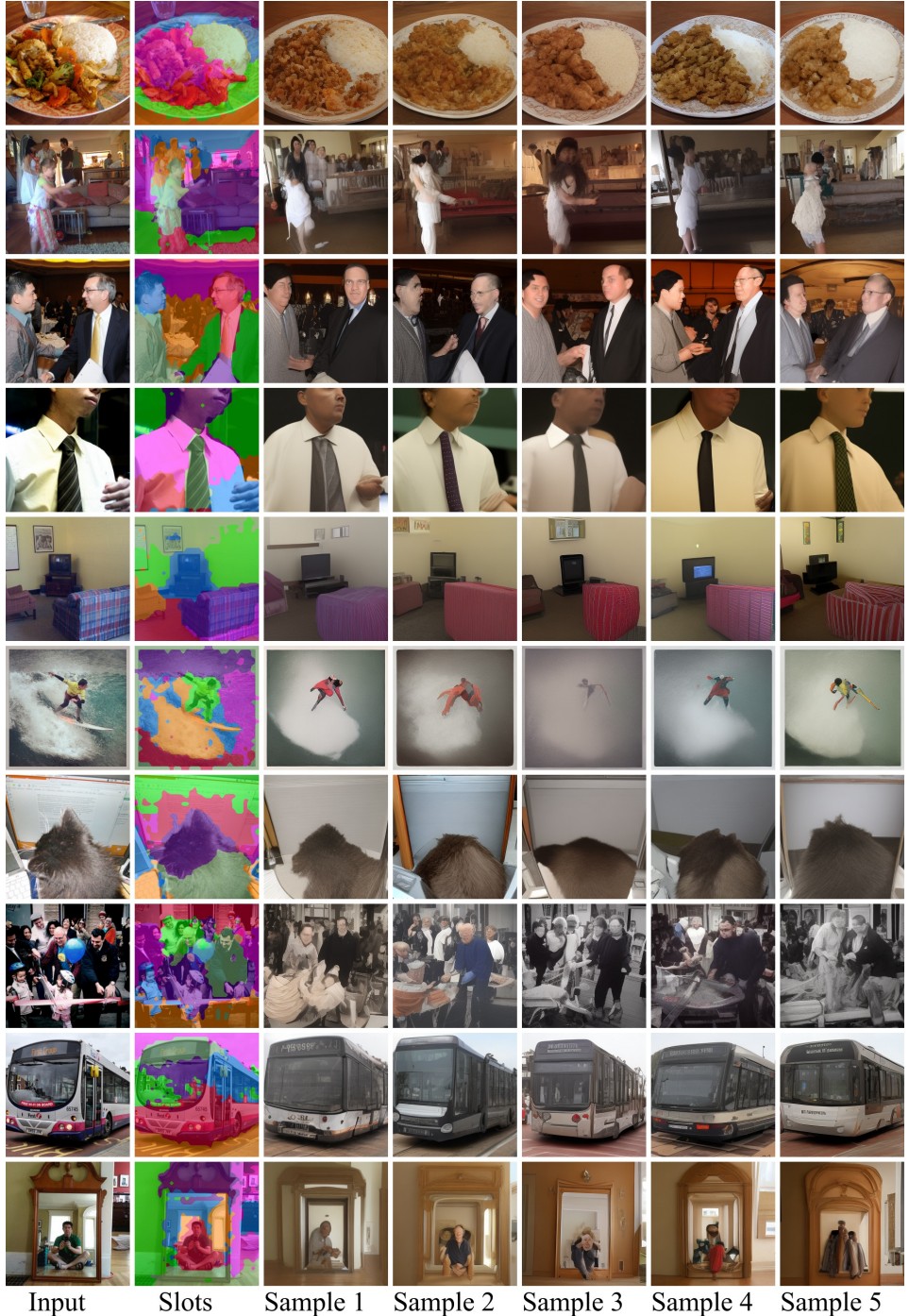

Input       Slots       Sample 1    Sample 2    Sample 3    Sample 4    Sample 5

Figure 23: **Random Samples for Real-World Image Generation 3.**

