# OpenReview forum: "Object-Centric Slot Diffusion"
_NeurIPS.cc/2023/Conference — NeurIPS 2023 spotlight_

### Official Review · Reviewer_QNuS · 2023-06-08

**Soundness:** 3 good
**Presentation:** 4 excellent
**Contribution:** 3 good
**Rating:** 7
**Confidence:** 5

**Summary:**

This paper proposes Latent Slot Diffusion (LSD), an object-centric learning framework combining the Slot Attention module and a Latent Diffusion Model (LDM) based slot decoder. The model is trained in an auto-encoding manner, where the loss is the slot-conditioned denoising loss in the LDM-based decoder. Extensive experiments demonstrate the effectiveness of LSD in 1) unsupervised scene decomposition  2) object property prediction  3) image generation/editing, surpassing previous SOTA SLATE which uses a auto-regressive Transformer-based slot decoder.

**Strengths:**

- The paper is well-written and easy to follow. The figures are of high quality
- The use of LDM as the slot decoder is very intuitive and reasonable, considering the recent trends in object-centric models (CNN decoder --> dVAE+Transformer decoder --> KL-VAE+LDM decoder). The results verify this design choice
- The experimental results cover lots of different tasks, and show very strong performance. The improvement in generation capacity is very impressive compared to baselines

**Weaknesses:**

I don't see any big issue with the paper. One might consider LSD as "simply replacing the slot decoder with an LDM", but I think this modification is reasonable, and works pretty well across datasets and tasks. SLATE/STEVE also just replace the CNN decoder in Slot Attention with Transformer decoders, and they have been widely used due to their strong unsupervised segmentation performance. I believe LSD's improvement in generation quality will also facilitate future research in object-centric generative models.

That being said, I think the paper should have more discussion of its limitations. See the `Questions` and `Limitations` sections below.

**Questions:**

1. Regarding the pre-trained VAE model:
- I appreciate the authors' efforts in having SLATE+ with the same VAE for a fair comparison. Have you tried training a VAE from scratch, and comparing this LSD variant with SLATE (which also trains dVAE from scratch)?

2. Regarding the LSD + Stable Diffusion (SD) experiments in the Appendix (**not affecting paper decision**):
- This is an interesting experiment, especially the results on the real-world COCO dataset. Do you have any numerical results to compare with the SOTA object-centric model DINOSAUR [1]?
- Can the authors show some generation or reconstruction results? I am curious because I believe even LSD cannot generate realistic unconstrained real-world images (FFHQ images are constrained as they only capture human faces). Will using a pre-trained SD decoder help here?

[1] Seitzer, Maximilian, et al. "Bridging the gap to real-world object-centric learning." ICLR. 2023.

3. Minor question (**not affecting paper decision**): a [concurrent work](https://arxiv.org/abs/2305.11281) you might want to cite, but I understand that paper comes out after the submission deadline.

**Limitations:**

The authors discuss limitations in Section 6. However, there are more limitations of this work (and the entire object-centric learning field):
- LSD is still unable to generate realistic unconstrained real-world images
- LSD suffers from the part-whole ambiguity issue, as can be seen from the COCO examples

I'd like to see the authors discuss these points in the paper.

---

> ### Author Rebuttal · Authors · 2023-08-10
>
> ### We genuinely appreciate your positive recommendations and insightful feedback!
>
> > I appreciate the authors' efforts in having SLATE+ with the same VAE for a fair comparison. Have you tried training a VAE from scratch, and comparing this LSD variant with SLATE (which also trains dVAE from scratch)?
>
> Thank you for this insightful suggestion! We agree that this experiment would provide valuable insights and we have plans to incorporate it into our final version.
>
> In our study, we chose to use the pretrained VAE instead of training a VAE from scratch, based on the following considerations:
>
> - The use of pre-trained VAEs is common practice when training diffusion models, primarily due to its training stability and demonstrated effectiveness.
> - Our experiments indicate that with a pretrained VAE, SLATE+ exhibits superior performance when compared to SLATE, which uses a VAE trained from scratch.
> - Furthermore, a single pretrained VAE demonstrated satisfactory performance across all four datasets we utilized. This suggests that the use of pretrained VAE can contribute to more efficient training processes.
>
> Given these reasons, we opt for using a pre-trained VAE in our approach.
>
> > Regarding the LSD + Stable Diffusion (SD) experiments in the Appendix (not affecting paper decision).
>
> > Do you have any numerical results to compare with the SOTA object-centric model DINOSAUR [1] (on COCO dataset)? Can the authors show some generation or reconstruction results? I am curious because I believe even LSD cannot generate realistic unconstrained real-world images (FFHQ images are constrained as they only capture human faces). Will using a pre-trained SD decoder help here?
>
> Thank you for the insightful suggestions. We agree that the recommended experiments would be significant additions to our work. In our tests, we do observe visual artifacts in decoded images when dealing with real-world scenes under the current (LSD+SD) model configuration. Such artifacts have constrained both the image reconstruction and generation capabilities of the model on the COCO dataset. However, we believe this is a very interesting direction worth dedicated effort, and we are diligently investigating this model design. If possible, we will include our findings including the quantitative results in our final revision.
>
> > Minor question (not affecting paper decision): a concurrent work (slotdiffusion) you might want to cite, but I understand that paper comes out after the submission deadline.
>
> Thank you for bringing this to our attention, we will include the discussion of the concurrent work in the revised paper.
>
> > The authors discuss limitations in Section 6. However, there are more limitations of this work (and the entire object-centric learning field):
> >
> > 1. LSD is still unable to generate realistic unconstrained real-world images
> > 2. LSD suffers from the part-whole ambiguity issue, as can be seen from the COCO examples
>
> Thank you for your valuable suggestion. We will be adding a dedicated limitation section in the revised version of our paper to provide a comprehensive explanation of these aspects.

---

> > ### Comment · Reviewer_QNuS · 2023-08-10
> > **Rebuttal Acknowledgment**
> >
> > I thank the authors for the response. I maintain my original rating of Accept after reading reviews from other reviewers.

---

### Official Review · Reviewer_ZjQw · 2023-07-01

**Soundness:** 3 good
**Presentation:** 3 good
**Contribution:** 3 good
**Rating:** 7
**Confidence:** 4

**Summary:**

Previous work has shown that transformer-based image generative models can be trained for object-centric learning which can handle complex scenes. That is, given unstructured observation, transformer-based models learn to find latent compositional structures to bind relevant features.

This work makes an attempt to explore the feasibility and potential of integrating diffusion models into object-centric learning.

A major contribution of this work is a novel Latent Slot Diffusion (LSD) model which combines conventional slot decoders with a conditional latent diffusion model, conditioned on object-centric slots provided by Slot Attention.

The authors have shown the effectiveness of the proposed model by evaluating several object-centric tasks including unsupervised object segmentation, downstream property prediction, compositional generation, and image editing.

**Strengths:**

- **Scope and relevance**:  Considering the growing interest in diffusion models, this paper is exceptionally timely as it broadens their application towards object-centric learning.

- **Significance of contributions**:  This paper presents a novel diffusion model for object-centric learning without the requirement of supervised annotations. It answers the raised question about how diffusion-based generative modeling can benefit object-centric learning properly.

- **Experimental results**:  The experiments in this paper are enough to prove the effectiveness of the proposed model.

- **Clarity**:  The main body of the paper is written very well.

**Weaknesses:**

- **Limited technical contribution**:  The proposed slot-contained diffusion is a trivial extension of existing text-to-image generation in Latent Diffusion Models (LDM) [59], which replaces text inputs with latent slots.

- **Unclear model details**:  To my understanding,  the whole model is trained for image generation objectives. How to apply such a generative model to unsupervised object segmentation is not clear.

- **Unclear training details**: if I am not mistaken, the object-centric encoder is trained independently rather than jointly with the latent slot diffusion decoder. I would suggest the authors clarify this. Thus, control experiments might be needed to study the effect of training strategies.

- **Some implementation details are missing**:  Without providing the source code, some important hyperparameters are not presented to justify the reproducibility of this work, eg., the number of $K$ for k-means cluster,  noise schedule $\alpha$, and steps $T$.

**Questions:**

In addition to the above weaknesses, here are two more questions:

- In line 231, given that the object segmentation masks are derived from the attention masks of Slot Attention, why using diffusion-based generative modeling can help unsupervised object segmentation?

- The LSD model has been evaluated across multiple tasks and outperforms existing state-of-the-art methods. However, it raises the question: to what extent does the enhanced performance truly stem from the diffusion-based generative modeling rather than the improved image encoder? It's an intriguing point to ponder.

Overall, this paper is a good effect. I will raise my rating if the authors can address my concerns.

**Limitations:**

The authors did not adequately address the limitations of this work.

A significant constraint of this study is the requirement for two distinct visual encoders: a pre-trained image auto-encoder and an object-centric encoder. This dual requirement results in a relatively heavy model for inference. This leads to the question: Is there any potential to share certain components between these two encoders to enhance efficiency?

---

> ### Author Rebuttal · Authors · 2023-08-10
>
> ### We sincerely appreciate for your constructive recommendation and valuable insights!
>
> > If I am not mistaken, the object-centric encoder is trained independently rather than jointly with the latent slot diffusion decoder. I would suggest the authors clarify this. Thus, control experiments might be needed to study the effect of training strategies.
>
> Thank you for your valuable feedback. We would like to clarify that the object-centric encoder is **trained jointly** with the diffusion decoder using the denoising loss. It is not trained independently.
>
> > The proposed slot-contained diffusion is a trivial extension of existing text-to-image generation in LDM, which replaces text inputs with latent slots.
>
> We appreciate your feedback and the opportunity to clarify our contributions. While the proposed model might seem like a combination of known components, the impact of our work exceeds the sum of its parts. In our model, the object-centric encoder (that provides the slots) and the diffusion decoder are **trained jointly**. This joint training has two important consequences:
>
> 1. **From an Object-Centric Learning Perspective**: Before our work, it was not known if using a diffusion decoder instead of transformer-based autoregressive decoders would lead to object discovery in the encoder. In this sense, it is remarkable that our model actually surpasses transformer-based autoregressive decoders—the current state-of-the-art in unsupervised scene decomposition. The importance of this finding is also resonated by reviewer QNuS.
> 2. **From a Diffusion Model Perspective**: Prior to our work, compositionality in diffusion models was achieved via text annotations. We show for the first time that compositionality can emerge in diffusion models without requiring text—solely via unsupervised training—allowing us to compositionally generate and edit scenes without text. This also points to the potential of harnessing vast amounts of unlabelled image data as a future avenue.
>
> > To my understanding, the whole model is trained for image generation objectives. How to apply such a generative model to unsupervised object segmentation is not clear.
>
> Thank you for this question. The model is trained with an image reconstruction objective, however, the learning signal is back-propagated through both the slot attention encoder and the decoder. The attention masks of slot attention emerging from this training process serve as unsupervised object segmentation.
>
> Specifically, in the encoder, the slots attend to a grid of image features. The area each slot attends is considered an object segment. This method for obtaining $\mathbf{A}$ is already detailed in Section 2.1. Please let us know what we might have overlooked; we're happy to provide additional details.
>
> > Some implementation details are missing: Without providing the source code, some important hyperparameters are not presented to justify the reproducibility of this work, eg., the number of for k-means cluster, noise schedule, and steps.
>
> We appreciate your feedback and attention to detail. While we have included the implementation details in the appendix of the paper, we acknowledge that there was an oversight regarding the number of $k$-means clusters. We will thoroughly review our implementation and documentation to ensure that all information is appropriately included. Additionally, we are committed to releasing our complete implementation upon acceptance.
>
> > In line 231, given that the object segmentation masks are derived from the attention masks of Slot Attention, why using diffusion-based generative modeling can help unsupervised object segmentation?
>
> Thank you for sharing the concern. As we clarified above, the attention masks are learned through a joint training of the object encoder module and the diffusion decoder using only the denoising loss. In our study, we have found that by combining these two components, the model naturally develops object segmentation capabilities without needing a supervision signal.
>
> > The LSD model has been evaluated across multiple tasks and outperforms existing state-of-the-art methods. However, it raises the question: to what extent does the enhanced performance truly stem from the diffusion-based generative modeling rather than the improved image encoder? It's an intriguing point to ponder.
>
> To answer this question, we have provided the comparison between LSD and SLATE+ in our study. This comparison essentially contrasts the performance of a transformer decoder versus a diffusion decoder. We believe the notable performance gap between LSD and SLATE+ does suggest that the diffusion decoder plays a key role in achieving the superior results.
>
> > The authors did not adequately address the limitations of this work.
>
> Thank you for your insightful suggestion. We will add a dedicated Limitation section in the revised manuscript to explain the limitations of our work.
>
> > A significant constraint of this study is the requirement for two distinct visual encoders: a pre-trained image auto-encoder and an object-centric encoder. This dual requirement results in a relatively hefty model for inference. This leads to the question: Is there potential to share certain components between these two encoders to enhance efficiency?
>
> Thank you for the insightful comment! One potential way to enhance efficiency is to share the encoder of the VAE with the object-centric encoder. In fact, as also discussed in our response to reviewer uRVS, we have investigated this specific configuration of LSD during our early experiments. However, our initial tests indicated suboptimal object segmentation and, as a result, we opted to keep the two components separate. We will include a discussion on this question in the revised version of the paper.
>
> Additionally, it is noteworthy that during inference, if the downstream tasks only require object segmentation or object representations, then only the object-centric encoder is required.

---

> > ### Comment · Reviewer_ZjQw · 2023-08-11
> > **Rebuttal Acknowledgment**
> >
> > I thank the reviewers for their detailed clarification, which has released my concerns about their work.  Therefore, I raise my rating to accept.

---

### Official Review · Reviewer_TTEN · 2023-07-03

**Soundness:** 3 good
**Presentation:** 3 good
**Contribution:** 3 good
**Rating:** 7
**Confidence:** 3

**Summary:**

The paper studies the diffusion models into object-centric learning. The authors introduce the concept of latent slot diffusion (LSD) that can replace slot decoders conditioned on object slots. The model can work in an unsupervised compositional mode without requiring annotations such as text. Their experiments show that LSD performs better in complex scenes comparing with other methods that do unsupervised compositional generation.

**Strengths:**

The proposed method LSD can be viewed as a model substituting conventional slot-decoders with a conditional latent diffusion model, where the conditioning is done via the object slot attentions. It can also be viewed as a unsupervised conditional compositional diffusion based generation. The ablation experiments are helping in explaining the crux.

**Weaknesses:**

Object segmentation using LDS is yet to be perfected, leading to suboptimal downstream applications.

**Questions:**

What are potential ways to improve segmentation using LSD?

**Limitations:**

As a generative model, the method needs to consider the privacy and impacts of image manipulation.

---

> ### Author Rebuttal · Authors · 2023-08-10
>
> ### We would like to express our appreciation to your insightful feedback!
> > Object segmentation using LDS is yet to be perfected, leading to suboptimal downstream applications.
>
> Thank you for your observation. While the object segmentation in LSD may have room for improvement, it's crucial to underscore that LSD is functioning under the **unsupervised learning setting**. The problem of unsupervised entity-level segmentation is a well-recognized challenge in both object-centric learning and computer vision, with no perfect solution yet. Nevertheless, our work shows significant progress compared to the previous state-of-the-art, which is demonstrated by the effectiveness of LSD in comparison with baseline models under this demanding condition.
>
> > What are potential ways to improve segmentation using LSD?
>
> Thank you for your question. We would like to share some potential ways to improve the segmentation performance of LSD as followings:
>
> - **Using post processing techniques to improve the resolution and boundary prediction of the mask.**
>     - As also discussed in our response to reviewer uRVS, improving the quality of segmentation masks might be achieved with post-processing techniques like bilateral solvers and conditional random fields. These methods utilize low-level features, such as RGB colors and pixel positions, to fine-tune the boundaries of the segmentation masks. Integrating this refinement process into our model can potentially lead to improved segmentation outcomes.
> - **Applying LSD on pre-trained diffusion models.**
>     - As mentioned in our paper's appendix, LSD's segmentation can further benefit from pre-trained diffusion models. Such models may alleviate the slots' burden to capture the perceptual details of the images, allowing them to concentrate more on object discovery and produce more accurate segmentation masks. We will provide further investigation of this direction in our revised appendix.
> - **Adding supervised signals from large scale segmentation datasets.**
>     - While LSD operates within unsupervised learning contexts in this study, one might consider shifting to a semi-supervised learning approach when object segmentation is a primary concern for the downstream application. Integrating supervised segmentation signals into the object-centric encoder for specific data samples has the potential to greatly improve segmentation accuracy.
>
> > Ethics Concerns: As a generative model, the method needs to consider the privacy and impacts of image manipulation.
>
> We acknowledge the importance of privacy issues and the impacts of the ability to perform image manipulation. We have discussed these issues and other potential social implications of LSD in the "Broader Impact" section in the appendix, and we will address additional ethical consideration in our revised version.

---

> > ### Comment · Reviewer_TTEN · 2023-08-10
> > **Post-rebuttal**
> >
> > Thank you for posting your rebuttal. I'll keep my original rating "accept".

---

### Official Review · Reviewer_uRVS · 2023-07-04

**Soundness:** 3 good
**Presentation:** 3 good
**Contribution:** 3 good
**Rating:** 8
**Confidence:** 5

**Summary:**

This work introduces a methodology for using diffusion-based models to obtain object-centric representations.
This is done using slot representations, obtained from the Slot Attention module applied to the original image, as a conditioning variable for a latent diffusion model. The diffusion model and the Slot Attention module are trained end-to-end.

The proposed model is called Latent Slot Diffusion (LSD) and is evaluated on unsupervised object segmentation, downstream property prediction, compositional image generation, and image editing. The datasets used for evaluation are ClevrTex, MOVi-E, MOVi-C and, for the first time in the field, FFHQ. The performance of LSD is evaluated quantitatively on the first 4 tasks and is compared against SLATE and SLATE+. The proposed model outperforms both baselines in the tasks according to most metrics.

**Strengths:**

**Originality**.
The paper proposes a novel way to combine two existing models: slot attention and latent diffusion. The presentation clearly shows what are the novel elements.

**Quality**.
The method shows a successful way to leverage diffusion in object-centric learning. Especially the results on property prediction show that it has concrete benefits for the representation learning itself, which might be useful for several different tasks, not only the ones shown in the paper.

**Clarity**.
The submission neatly shows all the experiments that were carried out and the description of the underlying method is clear.

**Significance**.
The work is a necessary exploration of leveraging the generative power of the diffusion approach in the object-centric setting, highlighting the complexities related to having too-strong of a decoder and providing clear examples of its strengths, especially by using a very complex dataset (FFHQ).

**Weaknesses:**

**Quality**.
The lack of error bars (i.e. standard deviation) in the analysis makes the quantitative analysis weaker. Additionally, it would be interesting to further explore the problems with FG-ARI, as it is currently the standard metric used in the field, and although I agree with the statements, this is only based on experience and intuition, and not a proper scientific analysis, which has not been carried out yet. Lack of comparison with traditional models such as the improved Slot Attention architecture proposed in [1] makes the performance of the model harder to evaluate. Are the good results obtained primarily due to the improved architecture (e.g. larger encoders, better decoder), or is there something fundamentally good about using diffusion (e.g. the iterative improvement) that results in better object representations?

**Clarity**
The description of the method used for unsupervised object segmentation is lacking. There is only a reference to the use of the attention masks from slot attention, it would be much better to refer directly to the nice mathematical notation used before in the text.

*References*
1) Biza, Ondrej, et al. "Invariant Slot Attention: Object Discovery with Slot-Centric Reference Frames." arXiv preprint arXiv:2302.04973 (2023).

**Questions:**

As object segmentation is performed using the masks from the slot attention module, it is clear that it is not possible to obtain masks with the full resolution of the images in many cases. Could this be an area of improvement for future work or is it possible to already try using some unsupervised super-resolution technique to get higher-quality masks? Could be worth considering object segmentation as a visualization of what the model is doing instead of a separate task?

Do you have any insights into how the model would perform if slot attention is applied to the latent representation of the latent diffusion model instead of the original image? This would help clarify the choices made during the development of the method.

**Limitations:**

The authors have sufficiently addressed the limitations of the proposed model, as well as the broader impact of their approach. However, this can be improved further, by being more explicit about direct effect of ethnic bias, ageism or other forms of mis-representation that the model could lead to, which is standard practice for modern diffusion model, considering the extreme high-quality of the generated images. Direct manipulation of certain characteristics enables these effects in a much easier way.

---

> ### Author Rebuttal · Authors · 2023-08-10
>
> ### We sincerely thank you for your positive recommendation and thoughtful comments!
> > ... lack of error bars
>
> We agree that incorporating error bars can better evaluate model’s robustness, and we intend to provide the results here. However, due to constraints in computing resources, we only managed to complete the MOVi-E experiment within the rebuttal window. The table below shows MOVi-E results with 3 seeds per model. The results demonstrate LSD’s consistent high performance, outperforming baseline models across all metrics. We hope the new analysis provides clear evidence of LSD’s strength and has properly addressed your concern. We will include the rest of the results in the revised version of the paper.
>
> |Segmentation|SLATE|SLATE$^+$|LSD (Ours)|
> |:-:|:-:|:-:|:-:|
> |mBO ($\uparrow$)|30.17 $\pm$ 2.09|22.17 $\pm$ 0.47|**38.96 $\pm$ 0.58**|
> |mIoU ($\uparrow$)|28.59 $\pm$ 2.03|20.63 $\pm$ 0.43|**37.64 $\pm$ 0.55**|
> |FG-ARI ($\uparrow$)| 46.06 $\pm$ 4.07|45.25 $\pm$ 0.94|**52.17 $\pm$ 1.09**|
>
> |Representation|SLATE|SLATE$^+$|LSD (Ours)|
> |:-:|:-:|:-:|:-:|
> |Position ($\downarrow$)|2.09 $\pm$ 0.15| 2.15 $\pm$ 0.09|**1.85 $\pm$ 0.06**|
> |3D B-Box ($\downarrow$)|3.36 $\pm$ 0.14| 3.37 $\pm$ 0.33|**2.94 $\pm$ 0.00**|
> |Category ($\uparrow$)|38.93 $\pm$ 0.20| 38.00 $\pm$ 0.45|**42.96 $\pm$ 0.26**|
>
> > ... would be interesting to further explore the problems with FG-ARI
>
> We appreciate that you agree with our observations. Previous studies [1,2] have also noted similar limitations of FG-ARI and recommended using additional metrics, e.g., mIoU. Therefore, in our experiments, we report 3 metrics: mIoU, mBO, FG-ARI; and our model has shown significant benefits in all of them. Since these are standard metrics in the line of object-centric learning, exploring their limitations is perhaps beyond the scope of this work. That being said, it is an interesting and important topic of future research.
>
> > ... comparison with traditional models such as the Invariant Slot Attention (ISA)
>
> Although we acknowledge that including a comparison to ISA would provide additional insights to our results, we do not consider it critical. Firstly, our study and the ISA paper approach different aspects. Our work focuses on improving representation learning and image generation quality, while ISA aims to obtain object representations that are invariant to position and scale. An interesting direction would be to combine our work with ISA to introduce invariance in the object representation. However, we leave this exploration to future studies.
>
> > ... importance of improved architecture vs diffusion process
>
> Thank you for your insightful question. We believe that the improvement in performance stems from a combination of both. The diffusion architecture (e.g., the UNet) utilized in LSD has been meticulously explored within the diffusion models literature. And the iterative denoising technique has also been adopted to achieve high image generation quality, as evident from the progression from DALL-E [3] to DALL-E 2 [4]. On the other hand, as highlighted in the SLATE paper and reaffirmed in our study, improved generation capacity can significantly improve the representation learning capability, particularly in scenarios involving complex scenes. Therefore, we hold the view that the combination of both architecture and the iterative process contributes to the observed improvement.
>
> > ... mathematical notation for unsupervised object segmentation
>
> We will clarify this in the revised version (e.g., in Appendix) using the notations in Section 2.1.
>
> > ... improvement to get higher-quality masks?
>
> We agree that it is a promising direction for future work. One reason for the reduced resolution is to reduce memory and computational costs. There are existing potential solutions to this challenge, such as using bilateral solvers or conditional random fields for post-processing. These methods incorporate low level features like RGB colors and pixel positions to refine the boundary of the scaled segmentation masks and have been effectively utilized in other works [5]. We believe they could potentially be integrated into LSD for even higher-quality masks. We will include this discussion in our revised manuscript.
>
> > ... considering object segmentation as a visualization of what the model is doing?
>
> We appreciate this perspective. The segmentation are derived from the cross-attention between the slot representation and image patches. It does serve as a visualization of the model's spatial disentangling process during inference of slot representations.
>
> > ... how the model would perform if slot attention is applied to the latent representation of the latent diffusion model?
>
> Yes, this particular configuration of LSD was explored in our preliminary stages. Initial experiments indicated that object segmentation was not optimal when sharing the VAE encoder's latents to learn the slots.
>
> Additionally, we note that with the VAE encoder shared with the object encoder, the segmentation resolution will be constrained by the VAE latent's resolution, which in the case of LSD, would be ⅛ of the original image size. We acknowledge the importance of clarifying this development choice and will be including a discussion on this investigation in the revised version.
>
> > ... limitations and broader impact
>
> Thank you for your valuable suggestion. We will be adding a dedicated limitation section in the revised version of our paper to provide a comprehensive explanation of these aspects.
>
> [1] Monnier, et al. "Unsupervised layered image decomposition into object prototypes." ICCV. 2021.
>
> [2] Zimmermann et al. "Sensitivity of Slot-Based Object-Centric Models to their Number of Slots." arXiv. 2023.
>
> [3] Ramesh, et al. "Zero-shot text-to-image generation." ICML. 2021.
>
> [4] Ramesh, et al. "Hierarchical text-conditional image generation with clip latents." arXiv. 2022.
>
> [5] Wang, et al. "Cut and learn for unsupervised object detection and instance segmentation." CVPR. 2023.

---

### Author Rebuttal · Authors · 2023-08-10

## **General Response**

We thank all reviewers for their insightful and positive feedback! We are encouraged that they find our work **novel** (uRVS, ZjQw), **timely** (ZjQw), and **both intuitive and nontrivial** (QNuS). They also highlighted its **practical implications** (uRVS) and **potential to facilitate future research** (QNuS). We are pleased that they recognized our empirical evaluation as **thoroughly conducted** (uRVS, QNuS, ZjQw), **demonstrating impressive improvement** (QNuS), and our paper **well-written and easy to follow** (uRVS, ZjQw, QNuS).

We would like to extend our gratitude to the ethics reviewers for their insights on the ethical dimensions of our work. We acknowledge the concerns about the image manipulation ability and are committed to addressing them explicitly in our revised manuscript.

We will respond to each reviewer’s concerns and questions separately below.

---

### Decision · Program_Chairs · 2023-09-21

**Decision:**

Accept (spotlight)

**Comment:**

Reviewers have come to a consensus on accepting this paper, the authors are expected to incorporate the suggestions from reviewers in the final camera ready version.